# The effectiveness of feature attribution methods and its correlation with automatic evaluation scores

**Giang Nguyen**[*]
Auburn University
nguyengiangbkhn@gmail.com

**Daeyoung Kim**
KAIST
kimd@kaist.ac.kr

**Anh Nguyen**[*]
Auburn University
anh.ng8@gmail.com

## Abstract

Explaining the decisions of an Artificial Intelligence (AI) model is increasingly critical in many real-world, high-stake applications. Hundreds of papers have either proposed new feature attribution methods, discussed or harnessed these tools in their work. However, despite humans being the target end-users, most attribution methods were only evaluated on proxy automatic-evaluation metrics [60, 78, 80]. In this paper, we conduct the first user study to measure attribution map effectiveness in assisting humans in ImageNet classification and Stanford Dogs fine-grained classification, and when an image is natural or adversarial (i.e. contains adversarial perturbations). Overall, feature attribution is surprisingly not more effective than showing humans nearest training-set examples. On a harder task of fine-grained dog categorization, presenting attribution maps to humans does not help, but instead hurts the performance of human-AI teams compared to AI alone. Importantly, we found automatic attribution-map evaluation measures to correlate poorly with the actual human-AI team performance. Our findings encourage the community to rigorously test their methods on the downstream human-in-the-loop applications and to rethink the existing evaluation metrics.

## 1 Introduction

Why did a computer vision system suspect that a person had breast cancer [77], or was an US capitol rioter [2], or a shoplifter [3]? The explanations for such high-stake predictions made by existing Artificial Intelligence (AI) agents can impact human lives in various aspects, from social [22], to scientific [54], and legal [3, 23, 31]. An image classifier's decisions can be explained via an *attribution map* (AM) [13], i.e. a heatmap that highlights the input pixels that are important for or against a predicted label. AMs can be useful in localizing tumors in x-rays [61], reflecting biases [41] and memory [56] of image classifiers, or teaching humans to distinguish butterflies [48].

Via AMs, part of the thought process of AIs is now transparent to humans, enabling human-AI collaboration opportunities. Can machines and humans work together to improve the accuracy of humans in image classification? This question remains largely unknown although, in other non-image domains, AMs have been found useful to humans in the downstream tasks [12, 39, 47, 66, 66].

Furthermore, it is also unknown whether any existing automatic evaluation scores may predict such effectiveness of AMs to humans, the target user of AMs. Most AMs were often only evaluated on

---

[*]Corresponding authors. Work done when GN was at KAIST.

35th Conference on Neural Information Processing Systems (NeurIPS 2021).

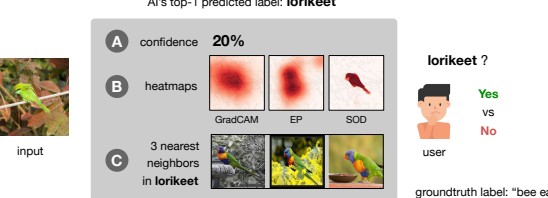

| Method | ImageNet | | Stanford Dogs | |
|---|---|---|---|---|
| | $\mu$ | $\sigma$ | $\mu$ | $\sigma$ |
| Confidence | 72.44 | 8.25 | **61.71** | 11.39 |
| GradCAM | 72.58 | 8.11 | 60.56 | 9.27 |
| EP | 73.85 | 6.88 | 56.67 | 10.57 |
| SOD | 72.06 | 7.63 | **61.67** | 10.87 |
| 3-NN | **76.08** | **5.86** | 57.20 | 10.58 |

(a) Given an input image, its top-1 predicted label (here, "lorikeet") and confidence score (A), we asked the user to decide Yes or No whether the predicted label is accurate (here, the correct answer is No)—i.e. the accuracy of human-AI teams *without* visual explanations. We also compared this baseline with the treatments where *one* attribution map (B) or a set of three nearest neighbors (C) is also provided to the user (in addition to the confidence score).

(b) Users of 3-NN outperform all other users including those using attribution maps on ImageNet classification. However, on fine-grained dog classification (Stanford Dogs), visual explanations (GradCAM, EP, 3-NN) surprisingly did not help but hurt the human performance. Results from testing humans on both real and adversarial images.

Figure 1: Overview of the methods compared in our user-study and its results.

proxy evaluation metrics such as pointing game [78], object localization [80], or deletion [60], which may not necessarily correlate with human-AI team accuracy on a downstream task.

In this paper, we conduct the first user-study to measure AM effectiveness in assisting humans on the 120-class Stanford Dog [38] fine-grained classification and 1000-class ImageNet [65] classification, which is the task that most attribution methods were designed for. We asked 320 lay and 11 expert users to decide whether machine decisions are correct after observing an input image, top-1 classification outputs, and explanations (Fig. 1a). This task captures two related quantities: (1) human accuracy on image categorization, here simplified to binary classification; and (2) model debugging, i.e. understanding whether AI is correct or wrong. For a fuller understanding of the effectiveness of AMs, we tested users on both real and adversarial images [75]. Our main findings include:[2]

1. AMs are, surprisingly, not more effective than nearest-neighbors (here, 3-NN) in improving human-AI team performance on both ImageNet and fine-grained dog classification (Sec. 3.2).

2. On fine-grained dog classification, a harder task for a human-AI team than 1000-class ImageNet, presenting AMs to humans interestingly does not help but instead hurts the performance of human-AI teams, compared to AI alone without humans (Fig. 2b).

3. On adversarial Stanford Dog images which are very hard for humans to label, presenting confidence scores only to humans is the most effective method compared to all other explanations tested including AMs and 3-NN examples (Sec. 3.2).

4. Despite being commonly used in the literature, some automatic evaluation metrics—here, Pointing Game [78], localization errors [80], and IoU [80]—correlate poorly with the actual human-AI team performance (Sec. 3.5).

5. According to a study on 11 AI-expert users, 3-NN is significantly more useful than Grad-CAM [67], a state-of-the-art attribution method for ImageNet classification (Sec. 3.6).

## 2   Methods

### 2.1   Image classification tasks

To evaluate human-AI team performance on image classification, for each image, we also presented to a user: (1) an AI's top-1 predicted label, (2) its confidence score, and, optionally (3) a visual explanation that is generated in attempt to explain the predicted label. The user was then asked to decide whether AI's predicted labels are correct or not (see Fig. 1a). We performed our experiments on two image datasets of ImageNet and Stanford Dogs, which are of increasing difficulty.

---

[2]Code and data are available at `http://github.com/anguyen8/effectiveness-attribution-maps`

**ImageNet**    We tested human-AI teams on ImageNet [65]—an image classification task that most attribution methods were tested on or designed for [7, 60, 67, 80]. ImageNet is a ∼1.3M-image dataset spanning across 1000 diverse categories from natural to man-made entities [65].

**Stanford Dogs**    To test whether our findings generalize to a fine-grained classification task that is harder to human-AI teams than ImageNet, we repeated the user-study on the 120-class Stanford Dogs [38] dataset, a subset of ImageNet. This dataset is challenging due to the similarity between dog species and the large intra-class variation [38]. Compared to ImageNet, Stanford Dogs is expected to be harder to human-AI teams because lay users often have significantly less prior knowledge about fine-grained dog breeds compared to a wide variety of everyday objects in ImageNet.

## 2.2    Image classifiers

We took ResNet-34 [34] pretrained on ImageNet (73.31% top-1 accuracy) from torchvision [50] as the target classifier (i.e., the AI) for both ImageNet and Stanford Dogs classification tasks because the 1000-class ImageNet that the model was pretrained on includes 120 Stanford Dogs classes. The visual explanations in this paper were generated to explain this model's predicted labels. We chose ResNet-34 because ResNets were widely used in feature attribution research [7, 28, 45, 60, 67].

## 2.3    Visual explanation methods

To understand the causal effect of adding humans in the loop, we compared the performance of a standard AI-only system (i.e. no humans involved) and human-AI teams. In each team, besides an input image and a corresponding top-1 label, humans are also presented with the corresponding confidence score from the classifier, and a visual explanation (i.e. one heatmap or three 3-NN images in; see examples from our user-study in Sec. A13).

**AI-only**    A common way to automating the process of accepting or rejecting a predicted label is via confidence-score thresholding [14]. That is, a top-1 predicted label is accepted if its associated confidence score is $\geq T$, a threshold. We found the optimal confidence threshold $T^*$ that produces the highest accepting accuracy on the entire validation set by sweeping across threshold values $\in \{0.05, 0.10, ..., 0.95\}$, i.e. at a 0.05 increment. The optimal $T^*$ values are 0.55 for ImageNet and 0.50 for Stanford Dogs (details in Sec. A6).

**Confidence scores only**    To understand the impact of visual explanations on human decisions, we also used a baseline where users are asked to make decisions given no explanations (i.e. given only the input image, its top-1 predicted label and confidence score). To our best knowledge, this baseline has not been studied in computer vision, but has been shown useful in some non-image domains for improving human-AI team accuracy or user's trust [12, 79].

**GradCAM and Extremal Perturbation (EP)**    We chose GradCAM [67] and Extremal Perturbation (EP) [28] as two representatives for state-of-the-art attribution methods (Fig. 1a). Representing for the class of white-box, *gradient*-based methods [16, 63, 80], GradCAM relies on gradients and the activation map at the last conv layer of ResNet-34 to compute a heatmap. GradCAM passed a weight-randomization sanity check [6] and often obtained competitive scores in proxy evaluation metrics (see Table 1 in [28] and Table 1&2 in [25]).

In contrast, EP is a representative for *perturbation*-based methods [7, 20, 29]. EP searches for a set of $a\%$ of input pixels that maximizes the target-class confidence score. We followed the authors' best hyperparameters—summing over four binary masks generated using $a \in \{2.5, 5, 10, 20\}$ and a Gaussian smoothing kernel with an std equal to 9% of the shorter side of the image (see the code [1]). We used the TorchRay package [5] to generate GradCAM and EP attribution maps.

We initially also tested vanilla Gradient [70], Integrated Gradient (IG) [74], and SHAP [46] methods in a pilot study but discarded them from our study since their AMs are too noisy to be useful to users in a pilot study.

**Salient object detection (SOD)**    To assess the need for heatmaps to explain a *specific classifer*'s decisions, we also considered a classifier-agnostic heatmap baseline. That is, we used a pre-trained state-of-the-art salient-object detection (SOD) method called PoolNet [43], which uses a ResNet-50

backbone pre-trained on ImageNet. PoolNet was trained to output a heatmap that highlights the *salient* object in an input image (Fig. 3)— a process that does *not* take our image classifier into account. Thus, SOD serves as a strong saliency baseline for GradCAM and EP.

**3-NN** To further understand the pros and cons of AMs, we compared them to a representative, prototype-based explanation method (e.g. [18, 53, 55]). That is, for a given input image $\mathbf{x}$ and a predicted label $\mathbf{y}$ (e.g., "lorikeet"), we show the top-3 nearest images to $\mathbf{x}$ by retrieving them from the same ImageNet training-set class $\mathbf{y}$ (Fig. 1a). To compute the distance between two images, we used the $L_2$ distance in the feature space of the last conv layer of ResNet-34 classifier (i.e. layer4 per PyTorch definition [4] or conv5_x in He et al. [34]). We chose layer4 as our pilot study found it to be the most effective among the four main conv layers (i.e. layer1 to layer4) of ResNet-34.

While $k$-NN has a wide spectrum of applications in machine learning and computer vision [68], the effectiveness of human-AI collaboration using prototype-based explanations has been rarely studied [64]. To the best of our knowledge, we provided the first user study that evaluates the effectiveness of post-hoc, nearest-neighbor explanations (here, 3-NN) on human-AI team performance.

## 2.4 User-study design

### 2.4.1 Participants

Our user-study experiments were designed and hosted on Gorilla [11]. We recruited lay participants via Prolific [57] (at $10.2/hr), which is known for a high-quality user base [59]. Each Prolific participant self-identified that English is their first language, which is the only demographic filter we used to select users on Prolific. Participants come from diverse geographic locations and backgrounds (see Sec. A9). Over the course of two small pilot studies and one main, large-scale study, we collected in total over 466 complete submissions (each per user) after discarding incomplete ones.

In the main study, after filtering out submissions by validation scores (described in Sec. 2.4.2), we had 161 and 159 *qualified* submissions for our ImageNet and Dogs experiments, respectively. Each of the 5 methods (i.e. except for AI-only) was experimented by at least 30 users, and each (image, explanation) pair was seen by at least two users (statistics of users and trials in Sec. A8).

### 2.4.2 Tasks

Our Gorilla study contains three main sets of screens: (1) Introduction—where each user is introduced to the study and relevant rules (details in Sec. A11); (2) Training; and (3) Test. Each user is randomly assigned a set of test images and *only one*[3] explanation method (e.g. GradCAM heatmaps) to work with during the entire study.

**Training** To familiarize with the image classification task (either ImageNet or Stanford Dogs), users were given five practice questions. After answering each question, they were shown feedback with groundtruth answers. In each training screen, we also described each component in a screen via annotations (see example screens in Sec. A11.2).

**Validation and Test** After training, each user was asked to answer 40 Yes/No questions in total. Before each question, a user was provided with a short WordNet [51] definition of the predicted label and three random training-set images (see an example in Fig. A5) correctly-classified into the predicted class with a confidence score $\geq 0.9$. To control the quality of user responses, out of 40 trials, we used 10 trials as validation cases where we carefully chose the input images such that we expected participants who followed our instructions to answer correctly (details in Sec. A11.3). We excluded those submissions that had below 10/10 and 8/10 accuracy on the 10 validation trials from the ImageNet and Dogs experiments, respectively. For the remaining (i.e., qualified) submissions, we used the results of their 30 non-validation trials in our study.

---

[3] Showing multiple explanation types to the same user complicates the user-training procedure and also makes the task harder to users due to context switching.

### 2.4.3 Images

We wish to understand the effectiveness of visual explanations when AIs are correct vs. wrong, and when AIs face real vs. adversarial examples [75]. For both ImageNet and Dogs experiments, we used the ResNet-34 classifier to sample three types of images: (1) correctly-classified, real images; (2) misclassified, real images; and (3) adversarial images (i.e. also misclassified). In total, we used 3 types × 150 images = 450 images per dataset. Each image was then used to generate model predictions and explanations for comparing the 6 methods described in Sec. 2.3.

**Filtering** From the 50K-image ImageNet validation set, we sampled images for both ImageNet and Dogs experiments. To minimize the impact of low-quality images to users' performance, we removed all 900 grayscale images and 897 images that have either width or height smaller than 224 px, leaving 48,203 and 5,881 images available for use in our ImageNet and Dogs experiments, respectively. For Dogs, we further excluded all 71 dog images mislabeled by ResNet-34 into non-dog categories (examples in Sec. A12) because they can trivialize explanations, yielding 5,810 Dogs images available for selection for the study.

**Selecting natural images** To understand human-AI team performance at varying levels of difficulty for humans, we randomly divided images from the pool of filtered, real images into three sets: Easy (E), Medium (M), and Hard (H).

Hard images are those correctly labeled by the classifier with a low confidence score (i.e., $\in [0, 0.3)$) and mislabeled with high confidence (i.e., $\in [0.8, 1.0]$). Vice versa, the Easy set contains those correctly labeled with high confidence and mislabeled with low confidence. The Medium set contains both correctly and incorrectly labeled images with a confidence score $\in [0.4, 0.6]$ i.e. when the AI is unsure (see confidence-score distributions in Sec. A7). In each set (E/M/H), we sampled 50 images correctly-labeled and 50 mislabeled by the model. In sum, per dataset, there are 300 natural images divided evenly into 6 *controlled* bins (see Fig. A1 for the ratios of these bins in the original datasets).

**Generating adversarial images** After the filtering above, we took the remaining real images to generate adversarial examples via Foolbox [62] for the ResNet-34 classifier using the Project Gradient Descent (PGD) framework [49] with an $L_\infty$ bound $\epsilon = 16/255$ for 40 steps, each of size of $1/30$. We chose this setup because at weaker attack settings, most adversarial images became *correctly* classified after being saved as a JPEG file [44], defeating the purpose of causing AIs to misbehave. Here, for each dataset, we randomly sampled 150 adversarial examples (e.g., the input image in Fig. 1a) that are *misclassified* at the time presented to users in the JPEG format and often contain so small artifacts that we assume to not bias human decisions. Following the natural-image sampling, we also divided the 150 adversarial images into three sets (E/M/H), each containing 50 images.

### 2.5 Automatic evaluation metrics for attribution maps

Many attribution methods have been published; however, most AMs were not tested on end-users but instead only assessed via proxy evaluation metrics. We aim to measure the correlation between three common metrics—Pointing Game [78], Intersection over Union (IoU) [80], and weakly-supervised localization (WSL) [80]—with the actual human-AI team performance in our user study.

All three metrics are based on the assumption that an AM for explaining a predicted label **y** should highlight an image region that overlaps with a human-annotated bounding box (BB) for that category **y**. For each of 300 real, correctly-classified images from each dataset (described in Sec. 2.4.3), we obtained its human-annotated BB from ILSVRC 2012 [65] for using in the three metrics.

**Pointing game** [78] is a common metrics often reported in the literature (e.g. [24, 28, 60, 63, 67, 76]). For an input image, a generated AM is considered a "correct" explanation (i.e. a *hit*) if its highest-intensity point lies inside the human-annotated BB. Otherwise, it is a *miss*. The accuracy for an explanation method is computed by averaging over all images. We used the TorchRay implementation of Pointing Game [5] and its default hyperparameters (tolerance = 15).

**Intersection over Union** The idea is to compute the agreement in Intersection over Union (IoU) [80] between a human-annotated BB and a binarized AM. A heatmap was binarized at the method's optimal threshold $\alpha$, which was found by sweeping across values $\in \{0.05x \mid x \in \mathbb{N} \text{ and } 0 < x < 20\}$.

**Weakly-supervised localization (WSL)** is based on IoU scores [80]. WSL counts a heatmap correct if its binarized version has a BB that overlaps with the human-labeled BB at an IoU $> 0.5$.

WSL is also commonly used e.g. [7, 24, 67]. For both GradCAM and EP, we found the binarization threshold $\alpha = 0.05$ corresponds to their best WSL scores on 150 correctly-classified images (i.e., excluding mislabeled images because human-annotated BBs are for the groundtruth labels).

# 3 Results

## 3.1 3-NN is more effective than AMs on ImageNet but both hurt user accuracy on Dogs

We measure the per-user accuracy on ImageNet classification over our controlled image sets (correctly-labeled, incorrectly-labeled, and adversarial images; see Sec. 2.4.3), i.e. on both natural and adversarial images. We repeated the same study for Stanford Dogs (see Table 1b).

**ImageNet**   We found that 3-NN is significantly better than SOD and GradCAM (Mann Whitney U-test; $p < 0.035$). The differences among feature attribution methods (GradCAM, SOD, EP) are *not* statistically significant. In terms of mean and standard deviation of per-user accuracy, users of 3-NN outperformed all other users by a large margin of ~3% (Table 1b; 76.08% and 5.86%).

**Stanford Dogs**   We found that users of 3-NN and EP performed statistically significantly worse than the users with no explanations (Table 1b; Confidence) and SOD users (Mann Whitney U-test; $p < 0.024$). There are no significant differences among other pairs of methods. That is, interestingly, on fine-grained Stanford Dog classification where an image may be real or adversarial, visual explanations (i.e. both 3-NN and AMs) *hurt* user classification accuracy.

## 3.2 On natural images, how effective are attribution maps in human-AI team image classification?

We wish to understand the effectiveness of AMs (GradCAM and EP) compared to four baselines (AI-only, Confidence, SOD, and 3-NN) in image classification by human-AI teams. Here, we compare 6 methods on 2 *natural*-image sets: ImageNet and Stanford Dogs (i.e. no adversarial images).

**Experiment**   Because a given image can be mapped to one of the 6 controlled bins as described in Sec. 2.4.3, for each of the 6 methods, we computed its human-AI team accuracy for each of the 6 bins where each bin has exactly 50 images (Sec. A1 reports per-bin accuracy scores). To estimate the overall accuracy of a method on the *original* ImageNet and Dogs dataset, we computed the weighted sum of its per-bin accuracy scores where the weights are the frequencies of images appearing in each bin in practice. For example, 70.93% of the Dogs images are correctly-classified with a high confidence score (Fig. A1; Easy Correct).

**ImageNet results**   On ImageNet, human-AI teams where humans use heatmaps (GradCAM, EP, or SOD) and confidence scores outperformed AI-only by an absolute gain of ~6–8% in accuracy (Fig. 2a; 80.79% vs. 88.77%). That is, when teaming up, **humans and AI together achieve a better performance than AI alone**. Interestingly, only half of such improvement can be attributed to the heatmap explanations (Fig. 2a; 84.79% vs. 88.77%). That is, users when presented with the input image and the classifier's top-1 label and confidence score already obtained +4% boost over AI alone (84.79% vs. 80.79%).

**Dogs results**   Interestingly, the trend did not carry over to fine-grained dog classification. On average, humans when presented with (1) confidence scores only or (2) confidence scores with heatmaps all *underperformed* the accuracy of AI-only (Fig. 2b; 81.14% vs. 76.45%). An explanation is that ImageNet contains ~50% of man-made entities, which contain many everyday objects that users are familiar with. Therefore, the human-AI teaming led to a substantial boost on ImageNet. In contrast, most lay users are not dog experts and therefore do not have the prior knowledge necessary to help them in dog identification, resulting in even worse performance than a trained AI (Fig. 2b; 81.14% vs. 76.45%). Interestingly, when providing users with nearest neighbors, human-AI teams with 3-NN outperformed all other methods (Fig. 2b; 82.88%) including the AI-only.

**Both datasets**   On both ImageNet and Dog distributions, **3-NN is among the most effective**. On average over 6 controlled bins, 3-NN also outperformed all other methods by a clear margin of 2.71% (Sec. A2 & A4). Interestingly, SOD users tend to reject more often than other users (Sec. A10), inadvertently causing a high human-AI team accuracy on AI-misclassified images (Sec. A14.5).

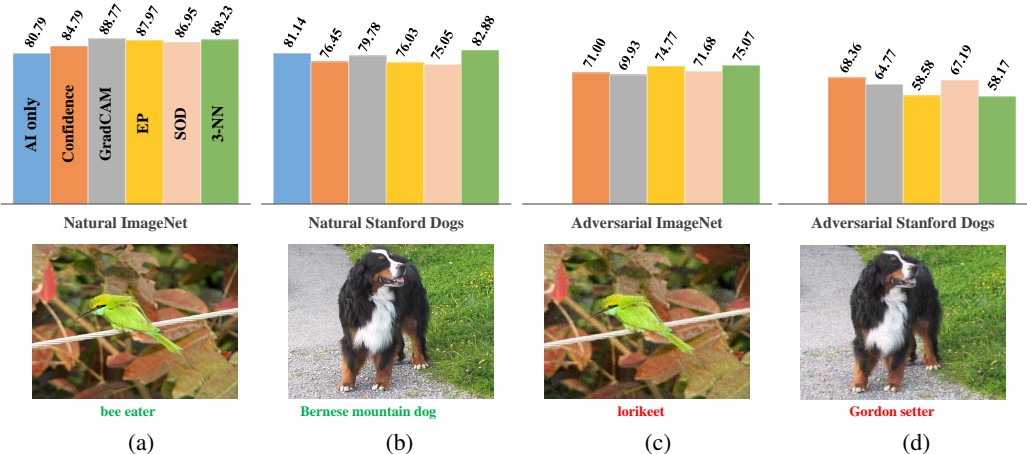

Figure 2: 3-NN is consistently among the most effective in improving human-AI team accuracy (%) on natural ImageNet images (a), natural Dog images (b), and adversarial ImageNet images (c). On the challenging adversarial Dogs images (d), using confidence scores only helps humans the most in detecting AI's mistakes compared to using confidence scores and one visual explanation. Below each sample dataset image is the top-1 predicted label (which was correct or wrong) from the classifier.

### 3.3 On adversarial images, how effective are attribution maps in human-AI team image classification?

As adversarial examples are posing a big threat to AI-only systems [8, 9, 52, 58], here, we are interested in understanding how effective explanations are in improving human-AI team performance over AI-only, which is assumed to be 0% accurate under adversarial attacks. That is, on ImageNet and Dogs, we compare the accuracy of human-AI teams on 150 adversarial examples, which all caused the classifier to misclassify. Note that a user was given a random mix of natural and adversarial test images and was not informed whether an input image is adversarial or not.

**Adversarial ImageNet** On both natural and adversarial ImageNet, AMs are on-par or more effective than showing confidence scores alone (Fig. 2c). Furthermore, the effect of 3-NN is a consistent +4% gain compared to Confidence (Fig. 2a & c; Confidence vs. 3-NN). Aligned with the results on natural ImageNet and Dogs, 3-NN remains the best method on adversarial ImageNet (Fig. 2c; 75.07%). See Sec. A14.4 for adversarial images that were only correctly-rejected by 3-NN users but not others.

**Adversarial Dogs** Interestingly, on adversarial Dogs, adding an explanation (either 3-NN or a heatmap) tend to cause users to agree with the model's incorrect decisions (Fig. 2d). That is, heatmaps often only highlight a coarse body region of a dog without pinpointing an exact feature that might be explanatory. Similarly, 3-NN often shows examples of an almost identical breed to the groundtruth (e.g. mountain dog vs. Gordon setter), which is hard for lay-users to tell apart (see qualitative examples in Sec. A14.6).

### 3.4 On ImageNet, why is 3-NN more effective than attribution maps?

Analyzing the breakdown accuracy of 3-NN in each controlled set of Easy, Medium, and Hard, we found 3-NN to be the most effective method in the Easy and Hard categories of **ImageNet** (Table A3).

**Easy** When AI mislabels with low confidence, 3-NN often presents contrast evidence showing that the predicted label is incorrect, i.e. the nearest examples from the predicted class is distinct from the input image (Fig. 1a; lorikeets have a distinct blue-orange pattern not found in bee eaters). The same explanation is our leading hypothesis for 3-NN's effectiveness on adversarial Easy, ImageNet images. More examples in Sec. A14.3.

**Hard** Hard images often contain occlusions (Figs. A9 & A12), unusual instances of common objects (Fig. A13), or could reasonably be in multiple categories (Figs. A11 & A10). When the classifier is *correct* but with low confidence, we found 3-NN to be helpful in providing extra evidence for users to

confirm AI's decisions (Fig. 3). In contrast, heatmaps often only highlight the main object regardless of whether AI is correct or not (Fig. 1a; GradCAM). More examples are in Sec. A14.1.

**Medium**   Interestingly, 3-NN was the best method on the Easy and Hard set, but not on the Medium (Table A3). Upon a closer look, we surprisingly found ∼63% of misclassified images by 3-NN human-AI teams to have debatable groundtruth ImageNet labels (see Sec. A14.2 for failure cases of 3-NN).

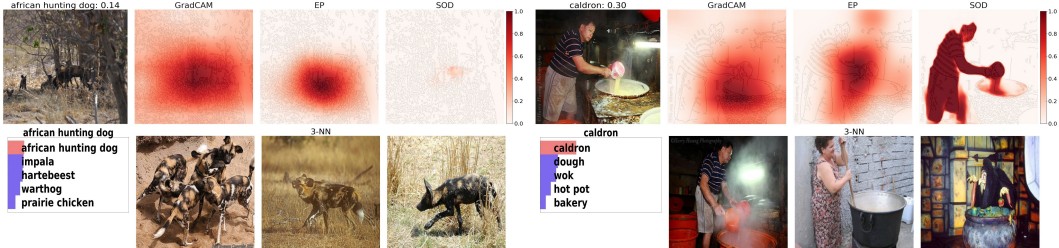

(a) "african hunting dog": dog-like mammals of Africa     (b) "caldron": a very large pot used for boiling

Figure 3: Hard images that were only correctly accepted by 3-NN users but not other users of GradCAM, EP, or SOD. Despite the animals in the input image are partially occluded, 3-NN provided closed-up examples of african hunting dogs, enabling users to correctly decide the label (a). Choosing a single label for a scene of multiple objects is challenging. However, 3-NN was able to retrieve a nearest example showing a very similar scene enabling users to accept AI's correct decisions (b). See full-res images in Figs. A9 & A10.

### 3.5   How do automatic evaluation scores correlate with human-AI team performance?

IoU [80], WSL [80], and Pointing Game [78] are three attribution-map evaluation metrics commonly used in the literature, e.g. in [7, 28, 29, 60, 67]. Here, we measure the correlation between these scores and the actual human-AI team accuracy to assess how high-performance on such benchmarks translate into the real performance on downstream tasks.

**Experiment**   We took the EP and GradCAM heatmaps of the 150 real images that were correctly-classified (see Sec. 2.4.3) from each dataset and computed their IoU, WSL, and Pointing Game scores[4] (see Sec. 2.5). We computed the Pearson correlation between these scores and the human-AI team accuracy obtained in the previous human-study (Sec. 2.4).

**Results**   While EP and GradCAM are state-of-the-art methods under Pointing Game [28] or WSL [29], we surprisingly found the accuracy of users when using these AMs to correlate poorly with the IoU, WSL, and Pointing Game scores of heatmaps. Only in the case of GradCAM heatmaps for ImageNet (Fig. 4a), the evaluation metrics showed a small positive correlation with human-AI team accuracy ($r = 0.22$ for IoU; $r = 0.15$ for WSL; and $r = 0.21$ for Pointing Game). In all other cases, i.e. GradCAM on Dogs; EP on ImageNet and Dogs (Fig. 4b–c), the correlation is negligible ($-0.12 \leq r \leq 0.07$). That is, the **pointing or localization accuracy of feature attribution methods do not necessarily reflect their effectiveness in helping users make correct decisions** in image classification.

### 3.6   Machine learning experts found Nearest-Neighbors more effective than GradCAM

We have found in our previous study on lay-users that AMs can be useful to human-AI teams, but not more than a simple 3-NN in most cases (Secs. 3.1–3.4). Here, we aim to evaluate whether such conclusions carry over to the case of machine learning (ML) expert-users who are familiar with feature attribution methods, nearest-neighbors, and ImageNet. Because ML experts have a deeper understanding into these algorithms than lay-users, our expert-study aims to measure the utility of these two visual explanation techniques to ML researchers and practitioners.

---

[4]Note that WSL and Pointing Game scores are binary while IoU scores are real-valued.

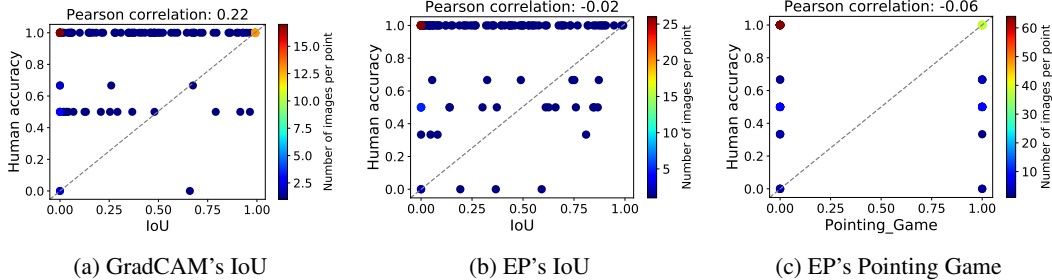

(a) GradCAM's IoU         (b) EP's IoU         (c) EP's Pointing Game

Figure 4: The ImageNet localization performance (here, under IoU vs. human-annotated bounding boxes) of GradCAM [67] and EP [28] attribution maps poorly correlate with the human-AI team accuracy (y-axis) when users use these heatmaps in image classification. Humans often can make correct decisions despite that heatmaps poorly localize the object (see the range 0.0–0.2 on x-axis).

As GradCAM was the best attribution method in the lay-user ImageNet study (Fig. 2), we chose GradCAM as the representative for attribution maps and compare it with 3-NN on human-AI team image classification on ImageNet.

**Experiment** We repeated the same lay-user study but on a set of 11 expert users who are Ph.D. students and postdoctoral researchers in ML and computer vision and from a range of academic institutions. We recruited five GradCAM users and six 3-NN users who are very familiar with feature attribution and nearest-neighbor algorithms, respectively. Similar to the lay-user study, each expert is presented with a set of randomly-chosen 30 images which include both natural and adversarial images.

**Results** The experts working with 3-NN performed substantially better than the GradCAM experts (Table 1; mean accuracy of 76.67% vs. 68.00%). 3-NN is consistently more effective on both natural and adversarial image subsets. Interestingly, the performance of GradCAM users also vary 3× more (standard deviation of 8.69% vs. 2.98% in accuracy). Aligned with the lay-user study, here, we found 3-NN to be more effective than AMs in improving human-AI team performance where the users are domain experts familiar with the mechanics of how explanations were generated.

Table 1: 3-NN is far more effective than GradCAM in human-AI team image classification of both natural and adversarial images. See the mean ($\mu$) and std ($\sigma$) in per-user accuracy over all images.

| | Users | Avg. validation accuracy | Natural | | Adversarial | | $\mu$ | $\sigma$ |
| --- | --- | --- | --- | --- | --- | --- | --- | --- |
| | | | Accuracy | Trials | Accuracy | Trials | | |
| GradCAM | 5 | 9.80/10 | 67.31 | 70/104 | 69.57 | 32/46 | 68.00 | 8.69 |
| 3-NN | 6 | 9.83/10 | **78.45** | 91/116 | **73.44** | 47/64 | **76.67** | **2.98** |

## 4 Related Work

**Attribution maps improved human accuracy in non-image domains** Our work is motivated by prior findings showing that AMs helped humans make more accurate decisions on non-image classification tasks including predicting book categories from text [66], movie-review sentiment analysis [12, 66], predicting hypoxemia-risk from medical tabular data [47], or detecting fake text reviews [39, 40]. In contrast, on house-price prediction, AMs showed marginal utility to users and even hurt their performance in some cases [21].

**Evaluating confidence scores on humans** AI confidence scores have been found to improve user's trust on AI's decisions [79] and be effective in human-AI team prediction accuracy on several NLP tasks [12]. In this work, we do not measure user trust but only the complementary human-AI team performance. To the best of our knowledge, our work is the first to perform such human-evaluation of AI confidence scores for image classification.

**Evaluating explanations on humans** In NLP, using attribution methods as a word-highlighter has been found to improve user performance in question answering [26]. Bansal et al. [12] found

that such human-AI team performance improvement was because the explanations tend to make users more likely to accept AI's decisions regardless of whether the AI is correct or not. We found consistent results that users with explanations tend to accept AI's predicted labels more (see Sec. A10) with the exception of SOD users who tend to reject more (possibly due to the prediction-agnosticity of SOD heatmaps).

In image classification, Chu et al. [19] found that presenting IG heatmaps to users did not significantly improve human-accuracy on predicting age from facial photos. Different from [19], our study tested multiple attribution methods (GradCAM, EP, SOD) and on the ImageNet classification task, which most attribution methods were evaluated on under proxy metrics (e.g. localization).

Shen and Huang [69] showed users GradCAM [67], EP [28], and SmoothGrad [72] heatmaps and measured user-performance in harnessing the heatmaps to identify a label that an AI incorrectly predicts. While they measured the effect of showing all three heatmaps to users, we compared each method separately (GradCAM vs. EP vs. SOD) in a larger study. Similar to [69], Alqaraawi et al. [10] and Folke et al. [27] tested the capability of attribution maps in helping users understand the decision-making process of AI. Our work differs from the above three papers [10, 27, 69] and similar studies in the text domain [15, 33] in that we did not only measure how well users understand AIs, but also the human-AI team accuracy on a standard *downstream* task of ImageNet.

Using AMs to debug models, Adebayo et al. [6] found that humans heavily rely on predicted labels rather than AMs, which is strongly aligned with our observations on the Natural ImageNet (in Sec. 3.2) where 3-NN outperforms AMs.

**Adversarial attacks** Concurrently, Folke et al. [27] also perform a related study that nearest-neighbor measures the effectiveness of both AMs and examples in helping users predict when AIs misclassify *natural adversarial* images [37], which we do not study in this paper. Our work largely differs from [27] in both the scale of the study, the type of image, and the experiment setup. To our knowledge, we are the first to evaluate human users on the well-known adversarial examples [49]. Previous work also found that it is easy to minutely change the input image such that the gradient-based AMs radically change [30]. However, none of such prior attempts studied the impact of their attacks to the actual end-users of the heatmaps.

# 5 Discussion and Conclusion

**Limitations** An inherent limitation of our study is that it is not possible to control the amount of prior knowledge that a participant has *before* entering the study. For example, a human with a strong dog expertise may perform better at the fine-grained dog classification. In that case, the utility of explanations is unknown in our study. We attempted to estimate the effect of prior knowledge to human-AI team accuracy by asking each user whether they know a class before each trial. We found prior knowledge to account for $\sim$1-6% in accuracy (Sec. A5). Due to COVID and the large participant pool, our study was done online; which, however, made it infeasible for us to control various physical factors (e.g. user performing other activities during the experiment) compared to a physical in-lab study. As most prior AM studies, we do consider state-of-the-art methods for training humans on explanations [71, 73] partly because each explanation method may need a different optimal training procedure.

To our knowledge, our work is the first to (1) evaluate human-AI team performance on the common ImageNet classification; (2) assess explanations on adversarial examples; (3) reveal the weak correlation between automatic evaluation metrics (Pointing Game, IoU [80], and WSL [80]) and the actual team performance. Such poor correlation encourages future interpretability research to take humans into their evaluation and to rethink the current automatic metrics [22, 42]. We also showed the first evidence in the literature that a simple 3-NN can outperform existing attribution maps, suggesting a combination of two explanation types may be useful to explore in future work. The superiority of 3-NN also suggests prototypical [17, 32] and visually-grounded explanations [35, 36] may be more effective than heatmaps.

## Acknowledgement

This research was supported by the MSIT Grant No. IITP-2021-2020-0-01489, and the Technology Innovation Program of MOTIE Grant No. 2000682. AN was supported by the NSF Grant No. 1850117 and a donation from NaphCare Foundation.

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
