# Appendix for:
# The effectiveness of feature attribution methods and its correlation with automatic evaluation scores

## A1  Break-down human-AI team accuracy on image subsets

In Table A1, for each subset mentioned in Sec. 2.4.3, we let $Acc_{B,P}$ denote the accuracy on the subset, in which $B$ is the bin (**E**, **M**, **H**) and $P$ is the prediction result (**C**orrect or **W**rong) produced by ResNet-34.

Table A1: Human-AI team accuracy (%) on **natural** images in 6 controlled bins. $Acc_{B,P}$.

| | $Acc_{B,P}$ **ImageNet** | | | | | | $Acc_{B,P}$ **Stanford Dogs** | | | | | |
|---|---|---|---|---|---|---|---|---|---|---|---|---|
| | **E** | | **M** | | **H** | | **E** | | **M** | | **H** | |
| | Correct | Wrong | Correct | Wrong | Correct | Wrong | Correct | Wrong | Correct | Wrong | Correct | Wrong |
| Confidence | 92.23 | 78.64 | 88.78 | 59.22 | 75.51 | 43.16 | 86.67 | 66.96 | 74.07 | 32.77 | 64.41 | 25.86 |
| GradCAM | **98.02** | 77.57 | **90.38** | **60.38** | 78.79 | 40.19 | 90.29 | 63.11 | **84.04** | 34.62 | 59.41 | 19.59 |
| EP | 97.14 | 86.41 | 84.11 | 58.18 | 78.64 | 37.38 | 87.63 | 58.88 | 79.81 | 22.22 | **70.64** | 15.24 |
| SOD | 95.69 | 82.57 | 88.60 | 55.65 | 66.96 | 43.75 | 83.96 | **70.19** | 70.18 | **40.78** | 59.43 | **28.04** |
| 3-NN | 96.64 | **88.60** | 86.84 | 54.95 | **83.93** | **44.76** | **97.46** | 56.00 | 76.92 | 24.55 | 60.42 | 18.75 |

**Controlled-to-original accuracy conversion**   While our experiments were not conducted on the real distributions of ImageNet and Dogs, we can still estimate the human accuracy of the explanation methods on the original datasets using the real ratios of subsets ($R_{B,P}$) presented in Fig. A1.

From Table A1 and Fig. A1, we measured human accuracy of explanation methods over the original distributions by the following formula:

$$\sum_{B \in \{E,M,H\}; P \in \{C,W\}} R_{B,P} \cdot Acc_{B,P}$$

The estimated accuracy was reported in Fig. 2. It should be noted that the above conversion was applied for natural images only because adversarial images were synthetically generated, so no original distribution exists.

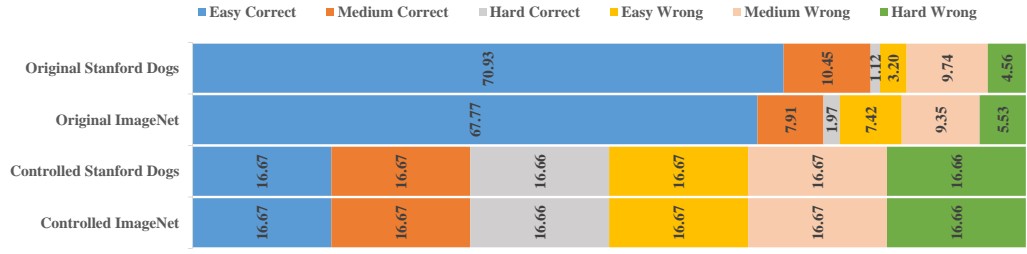

Figure A1: The ratios of image bins ($R_{B,P}$) in the controlled and real distributions.

## A2 How effective attribution maps in improving human-AI team accuracy on the controlled distribution?

On average, over the controlled distribution (i.e. 6 bins, each of 50 images), we found that 3-NN is the best method to help end-users categorize ImageNet images. 3-NN outperforms other explanation methods by at least 2.71% (Table A2; 76.59%). However, in fine-grained dog classification, explanations seem to have low utility. All methods scored nearly the same and close to the random baseline (i.e. 50% accuracy).

Table A2: Human-AI team accuracy with explanation methods on **random images** of the controlled distribution. NNs is the most effective in ImageNet experiment, while showing explanations could decrease human accuracy in fine-grained dog classification (Dogs).

|  | ImageNet | Dogs |
|---|---|---|
|  | Natural | Natural |
| Confidence | 73.17 | 58.33 |
| GradCAM | 73.88 | 58.47 |
| EP | 73.39 | 55.72 |
| SOD | 72.25 | **58.91** |
| 3-NN | **76.59** | 56.73 |

## A3 Are attribution maps more effective than 3-NN in improving human accuracy on Easy, Medium, and Hard images of AI on the controlled distribution?

In Table A3, we found 3-NN defeating other attribution methods and other baselines by a wide margin on easy and hard images of ImageNet classification task. It surpassed the second best methods (EP on **E** and Confidence on **H**) in each image set by 3.13% on average.

Yet, 3-NN provided very little information for users working with the Dogs experiment. None of attribution methods performed well on easy and hard images of dogs. Because the differences among dog classes are minimal, only rightly pointing out discriminative features could help users.

When AI classifies an image with confidence score around 0.5 (medium images), GradCAM was the best explanation method on both ImageNet and Dogs data, achieving 75.24% and 58.08%, respectively. However, the usefulness of explanation methods for users with dog images was insignificant since the net improvements over the random choice baseline were minimal.

Table A3: Human accuracy with explanation methods on easy (**E**), medium (**H**), and hard (**H**) images of the **controlled** distributions.

|  | ImageNet | | | Dogs | | |
|---|---|---|---|---|---|---|
|  | **E** | **M** | **H** | **E** | **M** | **H** |
| Confidence | 85.44 | 73.63 | 59.59 | 77.02 | 52.42 | **45.30** |
| GradCAM | 87.50 | **75.24** | 58.74 | 76.70 | **58.08** | 39.90 |
| EP | 91.83 | 70.97 | 57.62 | 72.55 | 51.72 | 43.46 |
| SOD | 89.33 | 72.05 | 55.51 | 77.14 | 56.22 | 43.66 |
| 3-NN | **92.70** | 71.11 | **64.98** | **78.44** | 50.00 | 39.58 |
| $\mu$ | 89.36 | 72.60 | 59.29 | 76.37 | 53.69 | 42.38 |

## A4 Are attribution maps more effective than 3-NN in improving human accuracy on correct/wrong images of the controlled distribution?

Explanations increase the chance humans agree with AI's predictions in sentiment classification and question answering [12]. While we examined the human accuracy on correct and images of AI, it has been shown that explanations also encouraged humans to accept AI's predictions rather than reject in image classification.

In Table A4, regarding ImageNet images, 3-NN was the most useful method for AI error identification, at 63.33%. Besides, the explanations from 3-NN and GradCAM seemed to benefit equally to humans when AI classify correctly (89.28% and 89.14%, respectively).

EP and 3-NN did the best in explaining network's correct predictions on dog images (79.03% and 79.56%). We were surprised because all methods did not outperform the random choice approach (50%) to identify misclassifications of ResNet-34 on Dogs images.

Table A4: Human accuracy with explanation methods on correct and wrong images of the controlled distributions.

|  | ImageNet | | Dogs | |
|---|---|---|---|---|
|  | Correct | Wrong | Correct | Wrong |
| Confidence | 85.62 | 60.80 | 75.14 | 41.71 |
| GradCAM | 89.14 | 59.38 | 77.85 | 39.47 |
| EP | 86.67 | 60.31 | 79.03 | 32.48 |
| SOD | 83.77 | 60.42 | 71.17 | **46.18** |
| 3-NN | **89.28** | **63.33** | **79.56** | 33.01 |

## A5 How does prior knowledge affect human accuracy?

Although we tried to mitigate the effect of prior knowledge by giving users the definition and sample images of categories, we still would like to examine how differently users score when never-seen-before vs. already-known objects are presented.

We let $\Delta$ denote the gap between accuracy on *known* images vs. accuracy on *unknown images* within a method. Looking into Table A5, SOD was affected the most by prior knowledge compared to other explanation methods with $\Delta$ of 6.38% in ImageNet or 2.21% in Dogs. An explanation for the above observation is that because SOD is model-agnostic, it provides the least classifier-relevant information, and therefore humans would have to rely on the prior knowledge the most to make decisions.

Table A5: Human accuracy on known and unknown images of both **natural** and **adversarial** images.

| | ImageNet | | | | | Dogs | | | | |
|---|---|---|---|---|---|---|---|---|---|---|
| | Known | | Unknown | | $\Delta$ | Known | | Unknown | | $\Delta$ |
| | Accuracy | Trials | Accuracy | Trials | | Accuracy | Trials | Accuracy | Trials | |
| Confidence | 73.07 | 724 | 69.89 | 176 | 3.18 | 61.88 | 446 | 61.59 | 604 | 0.29 |
| GradCAM | 72.69 | 714 | 72.22 | 216 | 0.47 | 60.78 | 459 | 60.32 | 441 | 0.46 |
| EP | 74.11 | 734 | 73.01 | 226 | 1.10 | 55.39 | 334 | 57.38 | 596 | -1.84 |
| SOD | 73.37 | 811 | 66.99 | 209 | **6.38** | 60.52 | 468 | 62.73 | 492 | **-2.21** |
| 3-NN | 76.46 | 838 | 74.32 | 182 | 2.14 | 56.35 | 367 | 57.75 | 563 | -1.40 |

## A6 AI-only thresholds

Using confidence score to automate the tasks, we found the two optimal thresholds are 0.55 and 0.50 for ImageNet and Dogs, respectively (see Table A6). These threshold values were tuned on around 50K ImageNet and 6K Stanford Dogs images.

Table A6: Accuracy of AI-only on ImageNet and Dogs at different threshold values.

| $T$ | 0.05 | 0.10 | 0.15 | 0.20 | 0.25 | 0.30 | 0.35 | 0.40 | 0.45 | 0.50 | 0.55 | 0.60 | 0.65 | 0.70 | 0.75 | 0.80 | 0.85 | 0.90 | 0.95 |
|---|---|---|---|---|---|---|---|---|---|---|---|---|---|---|---|---|---|---|---|
| **ImageNet** | 73.40 | 73.74 | 74.47 | 75.47 | 76.66 | 77.75 | 78.66 | 79.44 | 80.11 | 80.56 | **80.79** | 80.59 | 79.93 | 79.01 | 77.85 | 76.31 | 74.12 | 70.65 | 64.83 |
| **Dogs** | 77.37 | 77.44 | 77.81 | 78.23 | 78.81 | 79.31 | 79.80 | 80.65 | 80.65 | **81.14** | 81.07 | 80.34 | 79.34 | 77.67 | 75.82 | 73.63 | 70.02 | 65.21 | 55.93 |

## A7 Original distributions of ImageNet and Stanford Dogs

Fig. A2 shows the original distributions of ImageNet and Dogs by confidence intervals. Although we skipped images having confidence in [0.3, 0.4) and [0.6, 0.8), the human accuracy on the remaining intervals can still represent that of the original distributions because the numbers of images in those intervals are not major.

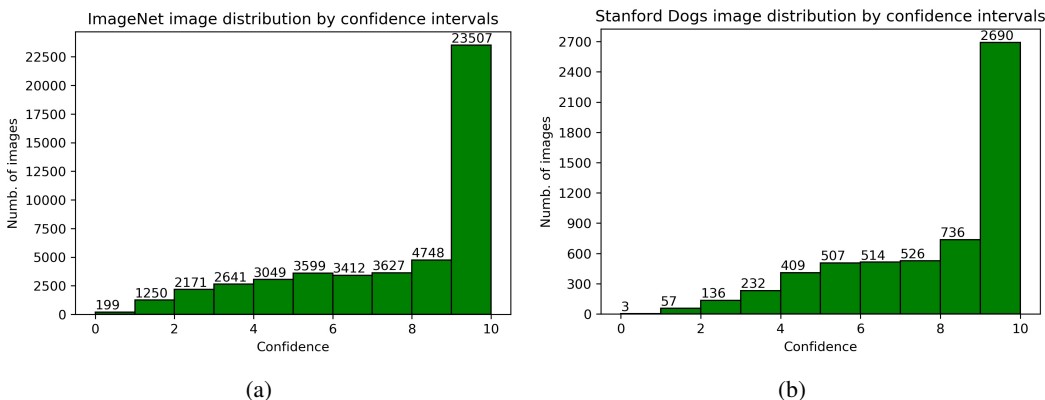

Figure A2: Image distribution of (a) ImageNet and (b) Dogs by confidence score intervals.

Table A7 shows the distributions of easy, medium, and hard images of ImageNet and Dogs dataset. It should be noted that we ignored a few confidence intervals so the total numbers of images are not 50K and 6K.

Table A7: The real distributions of easy, medium, and hard images in ImageNet and Dogs dataset.

| | **ImageNet** | | **Dogs** | |
|---|---|---|---|---|
| | Images | Percentage (%) | Images | Percentage (%) |
| **E** | 28967 | 75.19 | 3364 | 74.13 |
| **M** | 6648 | 17.26 | 916 | 20.19 |
| **H** | 2908 | 7.55 | 258 | 5.68 |
| $\sum$ | 38523 | 100 | 4538 | 100 |

## A8 Participants' statistics

Table A8 shows the numbers of users, trials, and users per image in ImageNet and Dogs experiment for each method.

Table A8: The number of users, trials, and users per image in ImageNet and Dogs experiment. A trial is one time we show an image to the user and asks for his decision.

|  | ImageNet | | | Dogs | | |
|---|---|---|---|---|---|---|
|  | Users | Trials | Users per image | Users | Trials | Users per image |
| Confidence | 30 | 900 | 2.00 | 35 | 1050 | 2.33 |
| GradCAM | 31 | 930 | 2.07 | 30 | 900 | 2.00 |
| EP | 32 | 960 | 2.13 | 31 | 930 | 2.07 |
| SOD | 34 | 1020 | 2.27 | 32 | 960 | 2.13 |
| 3-NN | 34 | 1020 | 2.27 | 31 | 930 | 2.07 |
| Total | **161** | **4830** |  | **159** | **4770** |  |

## A9 Participants' background and payment rate

**User background**   We only use a single criterion for filtering users: Users must be native English speakers. We think this is required for our study (and any study that uses English) since the training, instructions, and ImageNet labels are in English. We used this filter to avoid the cases where users make arbitrary decisions without understanding some words. Prolific shows that our users are diverse, aging from 18-77 (median=31) and coming from a diverse set of countries (US, UK, Poland, India, Korea, Canada, Australia, South Africa, etc.). Please see Prolific for more description of their online userbase, which, according to a study, is more reliable than AMT Turkers [59].

**Payment**   Our rate is higher than the Prolific recommended rate wage of $9.60/hr. In fact, during the study, we had increased our rate to attract more participants (up to $13.68/hr). As participants come from various countries in the world, we did not consider minimum wage per region because this recommended rate is suggested by Prolific and accepted by all participants.

## A10 Participants' acceptance-rejection rate

Table A9 shows the ratios of acceptance and rejection across methods in ImageNet and Dogs (natural examples only). Consistent with [12], we found that explanations increase the chance humans accept AI's predictions except SOD. As mentioned in Sec. A14.5, SOD users were most likely to reject AI's labels because this baseline gave users bad-quality heatmaps.

Table A9: The percentages of acceptance of rejection in ImageNet and Dogs experiment.

|  | ImageNet | | Dogs | |
|---|---|---|---|---|
|  | Accept | Reject | Accept | Reject |
| Confidence | 62.33 | 37.67 | 66.67 | 33.33 |
| GradCAM | **64.26** | 35.74 | 69.10 | 30.90 |
| EP | 62.99 | 37.01 | 73.27 | 26.73 |
| SOD | 61.97 | **38.03** | 62.66 | **37.34** |
| 3-NN | 63.56 | 36.44 | **73.40** | 26.60 |

## A11 Sample experiment screens

Here we show our experiment user interface by screens.

### A11.1 Instructions

We introduced Sam - the AI and explained the tasks to participants. Later, we explicitly restricted the device used for the experiments to ensure the display resolution will not affect human decisions.

Figure A3: Instruction screen of experiments.

## A11.2 Training

We gave users 5 trials where each of them is followed by a feedback screen on users' decision. For GradCAM, EP, and SOD, as their heatmaps have the similar format, the same interpretation was provided as shown in Fig. A4a. 3-NN displays to users three correct images of the predicted class in Fig. A4b, while Confidence only shows the input image, the predicted label, and the confidence score.

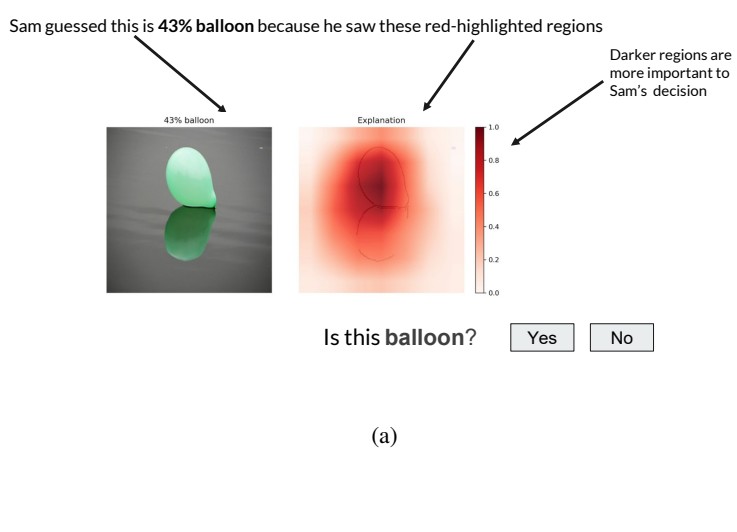

(a)

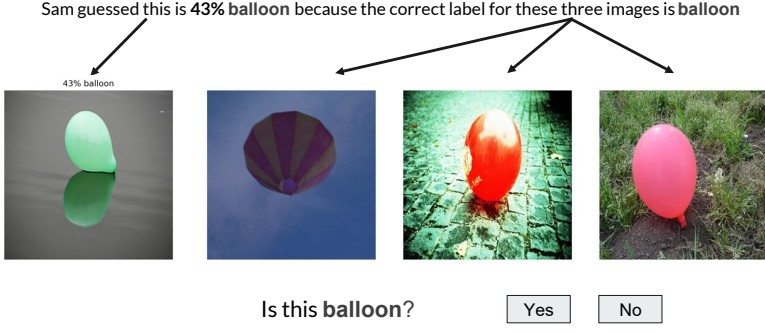

(b)

Figure A4: The training screens for (a) heatmaps (GradCAM, EP, and SOD) and (b) nearest neighbors with annotated components.

## A11.3 Validation and Test

While we have 10 trials in evaluation and 30 trials in test, we did not tell participants about the validation phase to avoid *overfitting*. These 40 trials were showed continuously in which the definition and sample images of the predicted class were given beforehand as in Fig. A5. No feedback was given in validation and test.

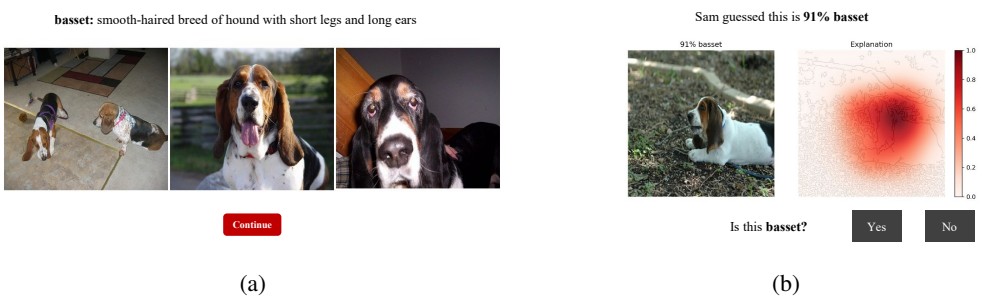

(a)                                    (b)

Figure A5: Users are given the definition and three sample images of the predicted classes (a). Users are asked to agree or disagree with the prediction of AI using explanation(b).

As the purpose of validation was to filter out users not paying enough attention to the experiments (e.g. random clicking), we carefully chose clearly wrong and correct images by ResNet-34 (Fig. A6) to check users' attention. It should be noted that the definition and sample images of the predicted category were given in advance.

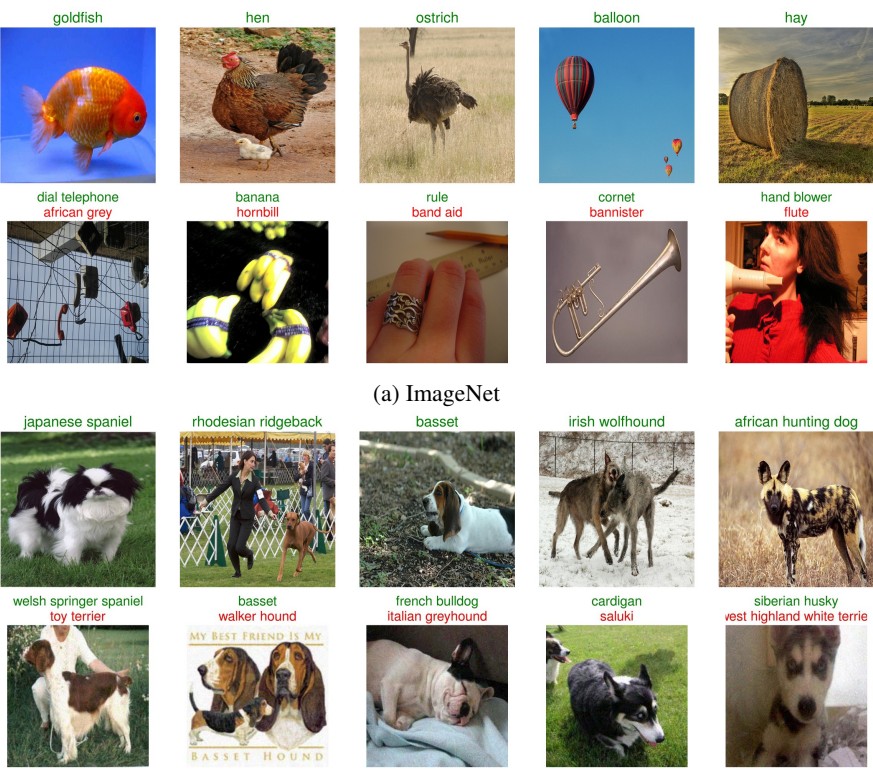

(a) ImageNet

(b) Stanford Dogs

Figure A6: Validation images for (a) ImageNet and (b) Dogs. There are 5 clearly correct and 5 clearly wrong predictions of AI in each experiment. Above each image is the groundtruth label and the misclassified label (if the image was misclassified by AI). In Stanford Dogs, the misclassified images were synthetically generated using PGD-$L_{\text{inf}}$ adversarial attacks.

## A12 Images mislabeled into non-dog categories (MIND samples)

In Dogs experiment, images misclassified into non-dog categories (MIND) were discarded because users often can instantly reject those images without explanation. As shown in Fig. A7, almost all MIND samples contain more than one object.

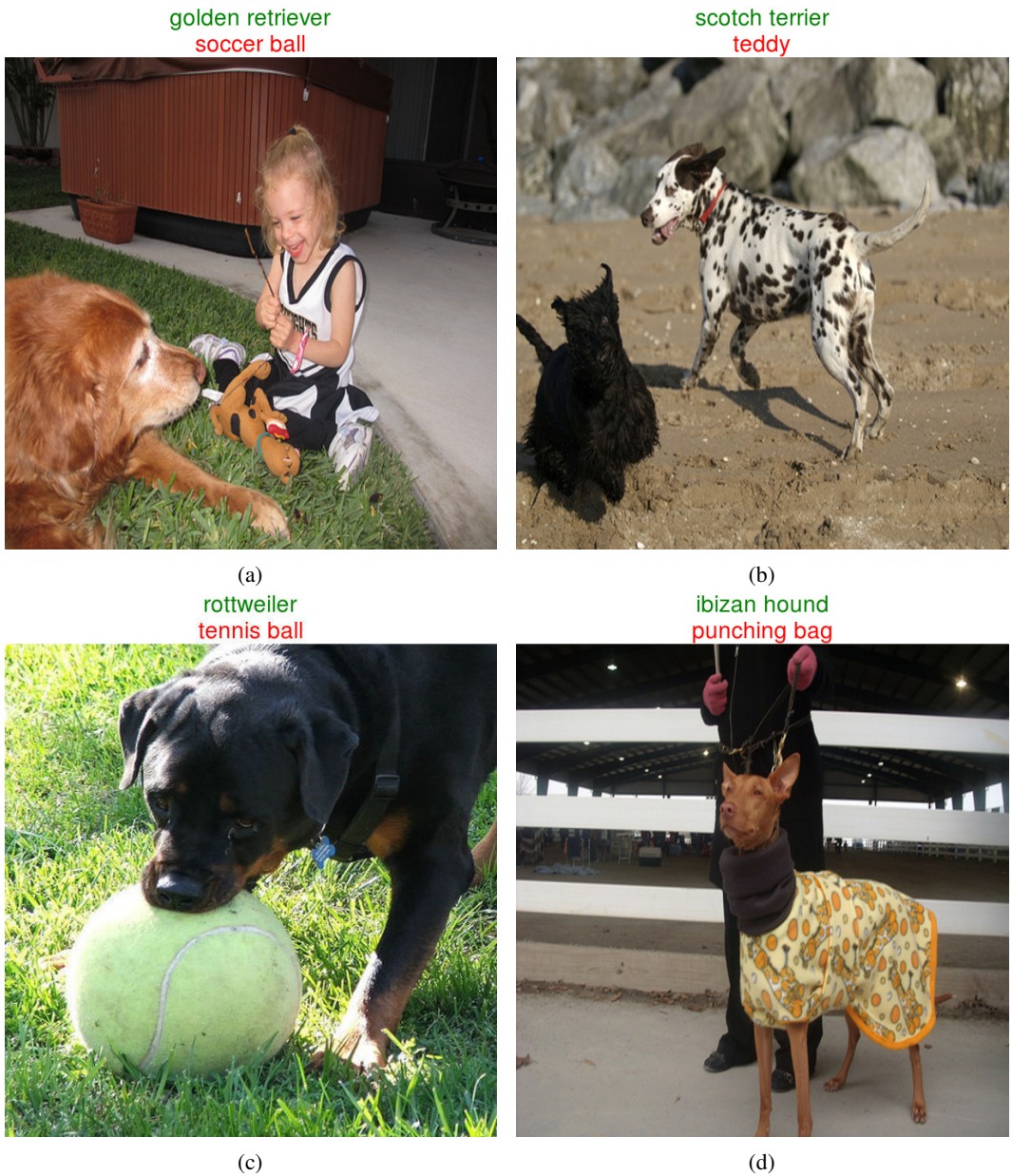

Figure A7: **M**islabeled **I**nto **N**on-**D**og samples (MINDs). There are 71 MINDs in Dogs by ResNet-34. Above each image is the groundtruth label and the misclassified label.

## A13 Example explanations displayed to users

Below are example of explanations taken from the screens displayed to users during our user-study. Here, we show the differences between GradCAM, EP, SOD, and 3-NN explanations for the same input image predicted as "american coot" (Fig. A8).

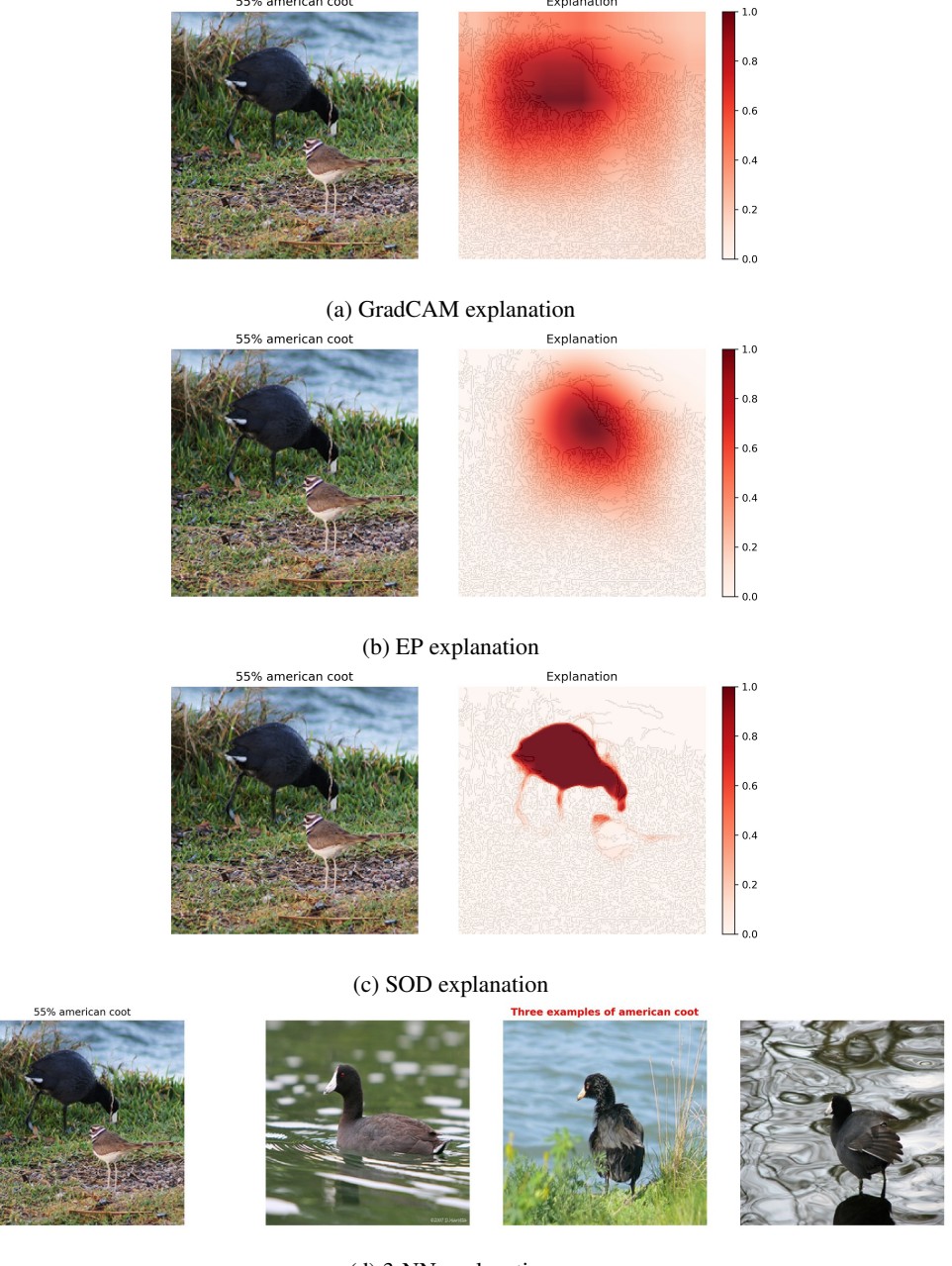

(a) GradCAM explanation

(b) EP explanation

(c) SOD explanation

(d) 3-NN explanation

Figure A8: The explanations from GradCAM, EP, SOD, and 3-NN for the input image labeled "american coot" by the classifier. While the highlight from GradCAM tends to be expansive, the focus of EP is narrow, and SOD attend to the entire body of the bird. 3-NN presents similar scenes of a coot around the pond.

## A14 Qualitative examples supporting our findings

### A14.1 Hard, real ImageNet images that were correctly labeled by 3-NN users but not GradCAM, EP, or SOD users

Regarding hard images, we observed that images corrected by 3-NN but not attribution maps and SOD often contain multiple concepts (Fig. A11), low quality (Fig. A9), look-alike objects (Fig. A10 and A14), only a part of the main object (Fig. A12), or objects with unusual appearances (Fig. A13). On these images, while heatmaps did not highlight the discriminative features and the confidence score was low, users tended to reject AI's labels. In contrast, 3-NN helped users gain confidence that AI is correct when there are multiple plausible labels.

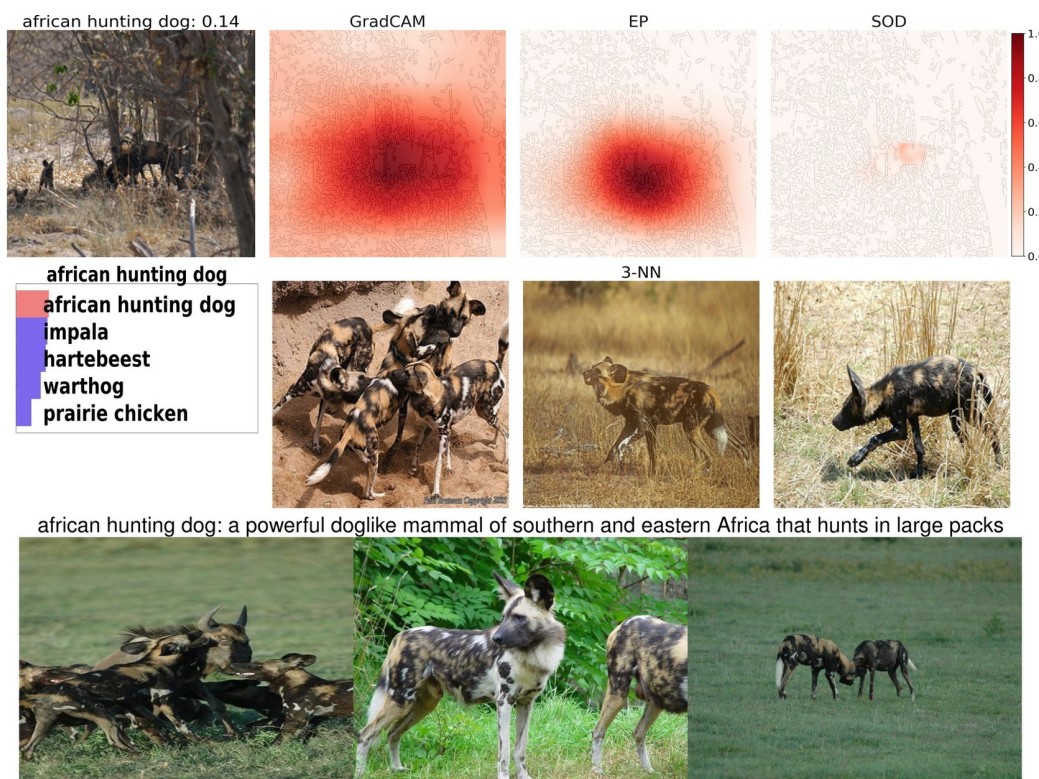

Figure A9: Hard ImageNet image with low quality. 3-NN might help users recognize the shape and color of "african hunting dog", while attribution methods gave users little information because users could not see what AI is looking at clearly.

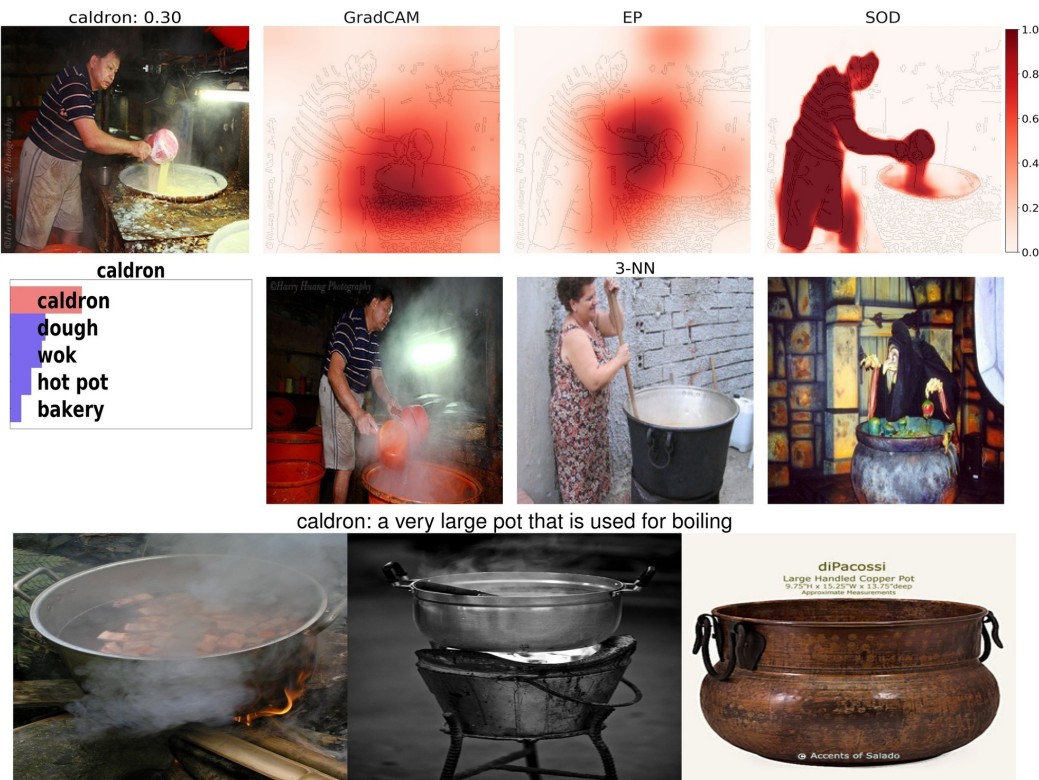

Figure A10: Hard ImageNet image with probable look-alike objects. 3-NN even found the same man, which strongly supports the AI prediction, while EP and SOD highlighted incorrectly.

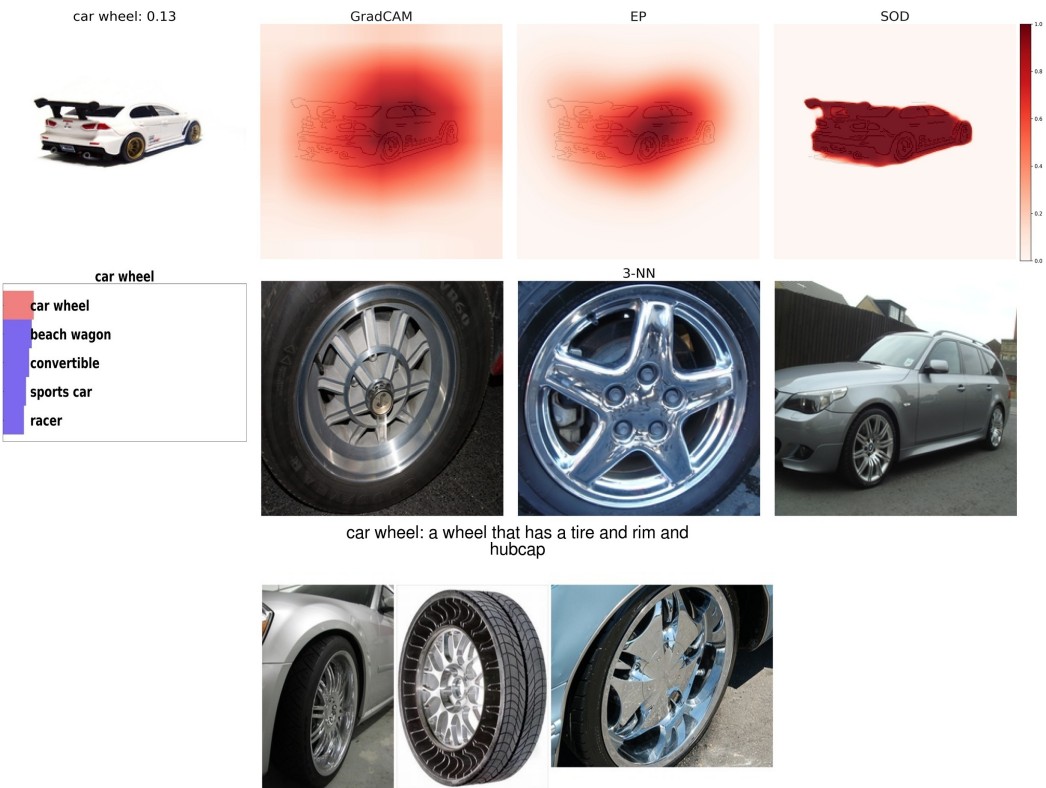

Figure A11: Hard ImageNet image with multiple plausible concepts present. The last image of 3-NN (right most) showed an image of a car labeled as "car wheel", which strongly helps users confirm the prediction of AI. However, users with other explanation methods rejected the prediction because AI did not highlight properly or showed low confidence.

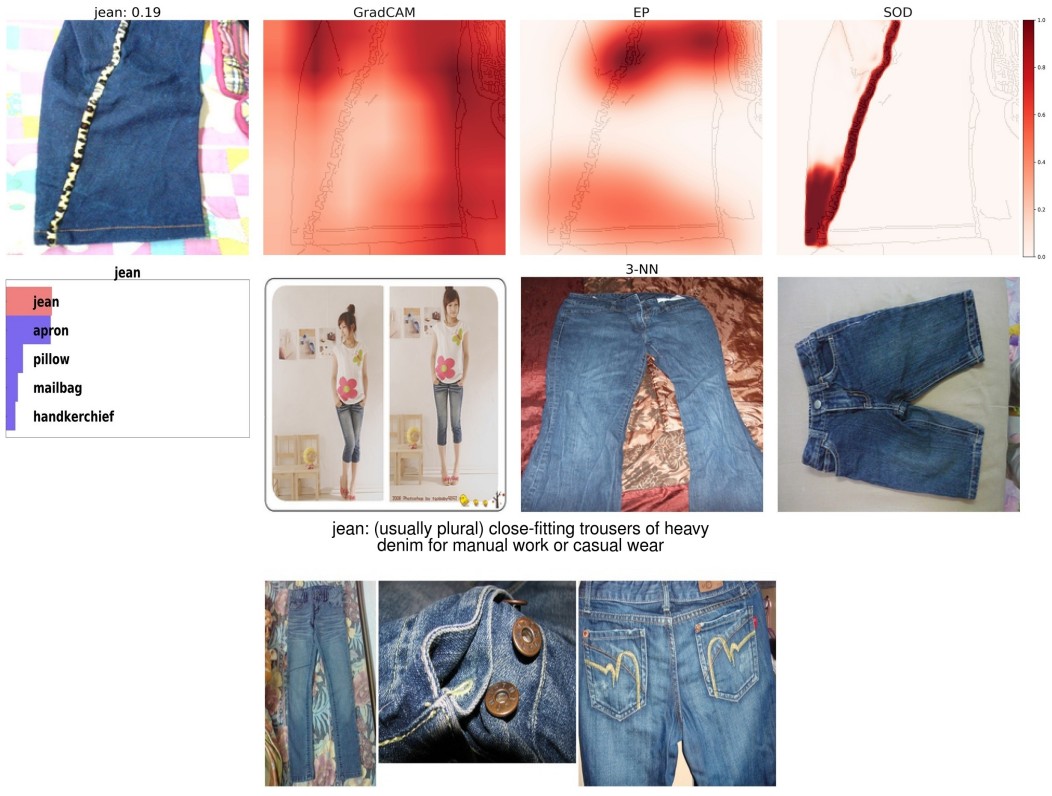

Figure A12: Hard ImageNet image with only a part of the main object. 3-NN helped users recognize a "jean" leg, but other heatmaps could not find the determinative features on the input image.

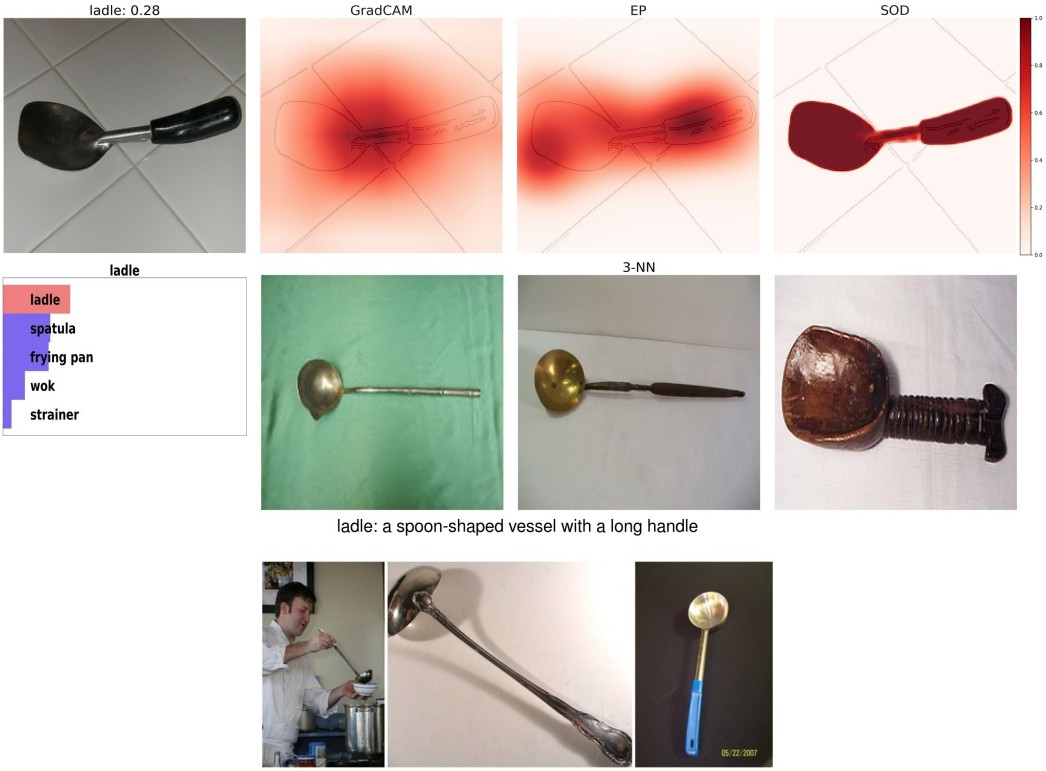

Figure A13: Hard ImageNet image with strange-looking objects. Although the "ladle" in the input image looks strange, 3-NN showed other ladles also have unusual appearance (the last neighbor image). Users might instantly rejected the prediction because the three sample images are contrastive.

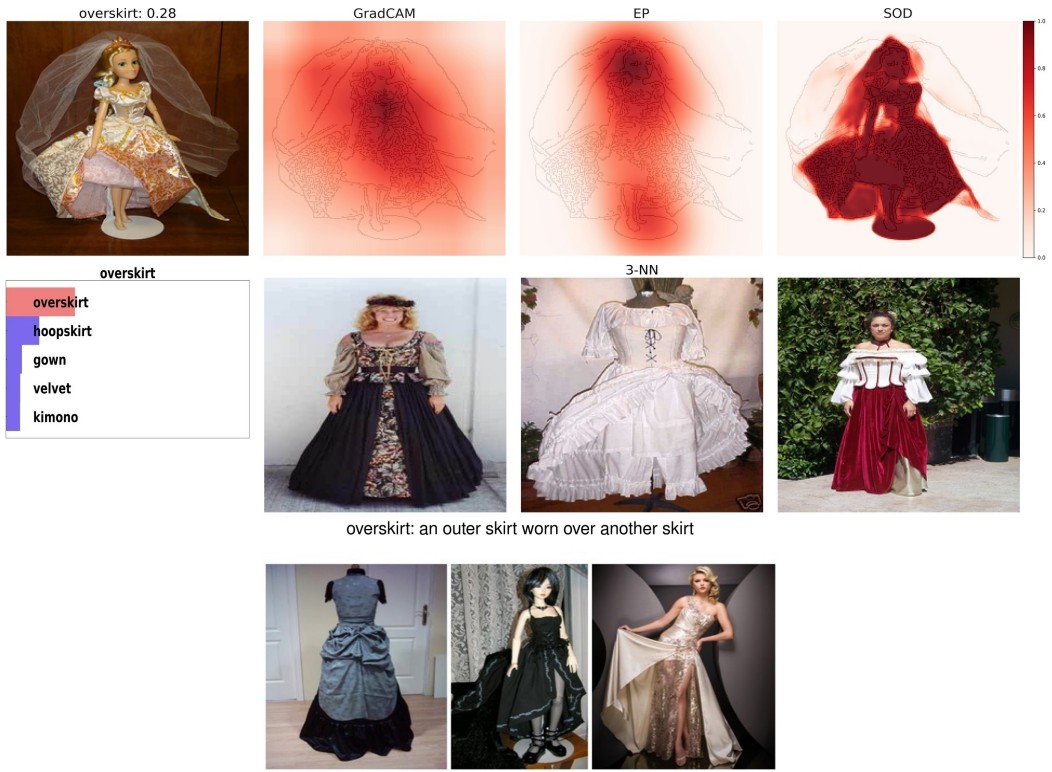

Figure A14: Hard ImageNet image with probable look-alike objects. 3-NN helped users gain confidence while "overskirt", "hoopskirt", and "gown" look very similar.

### A14.2 Medium, AI-misclassified, real ImageNet images that were incorrectly accepted by 3-NN users

These images can be divided into two main categories: debatable ground-truth (Figs. A15 & A16) and look-alike object (Figs. A17 & A18).

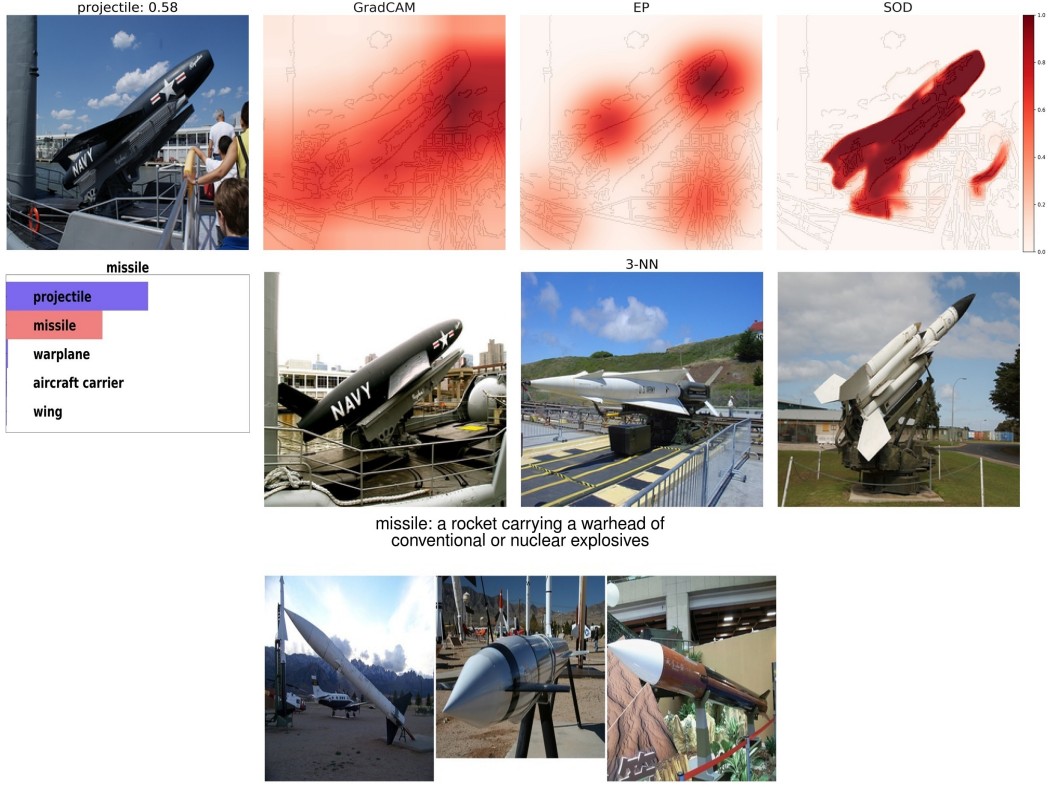

Figure A15: Medium ImageNet image with wrong labels. The input image and the first NN are clearly the same but the annotated label is "missile".

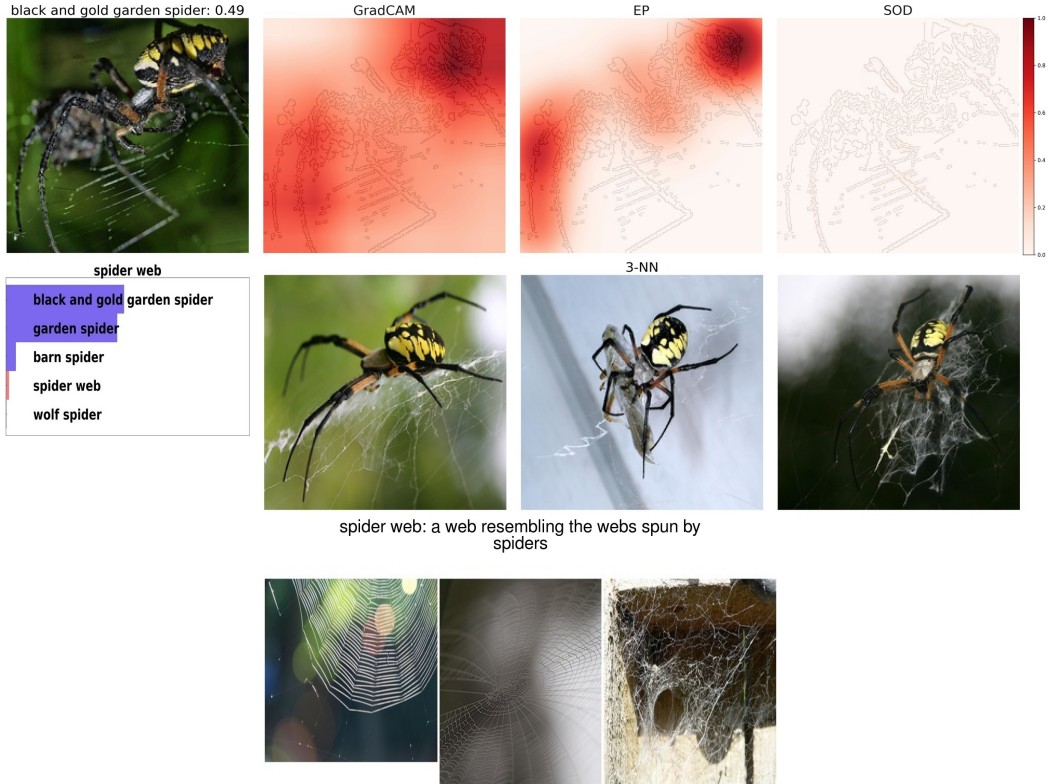

Figure A16: Medium ImageNet image with multiple objects present. The spider is salient and 3-NN retrieved very similar images, but the ground truth is "spider web".

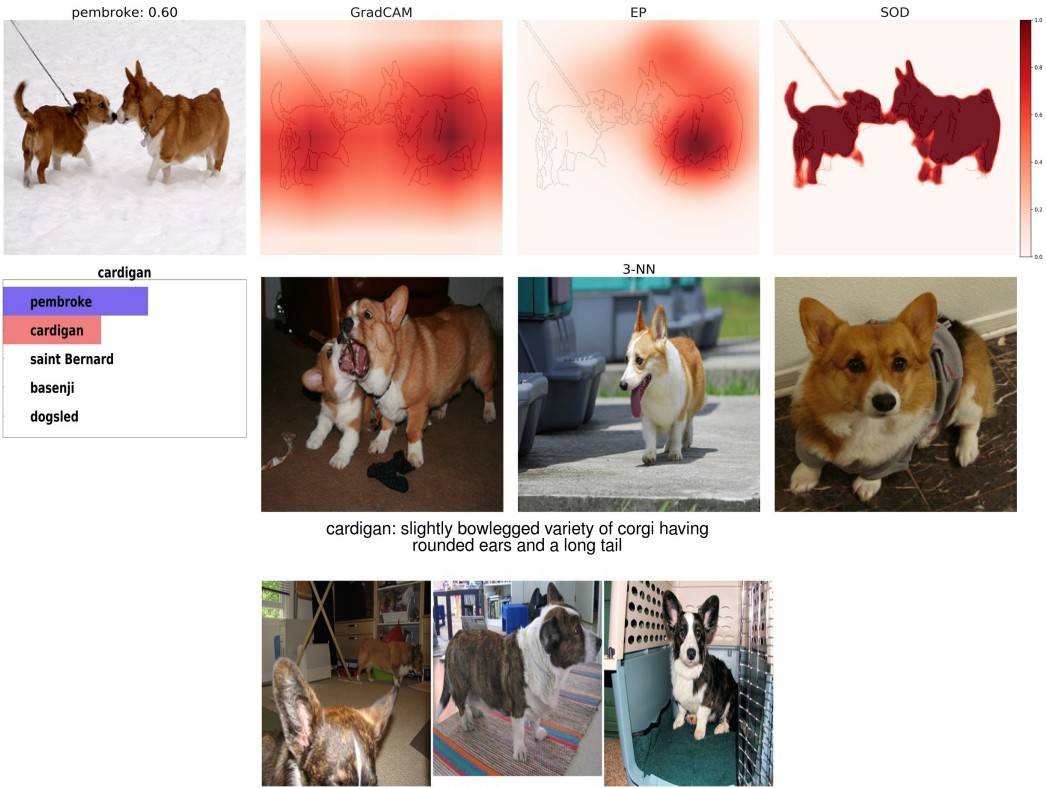

Figure A17: Medium ImageNet image with fine-grained classes. 3-NN failed to show the difference between "cardigan" and "pembroke" to users.

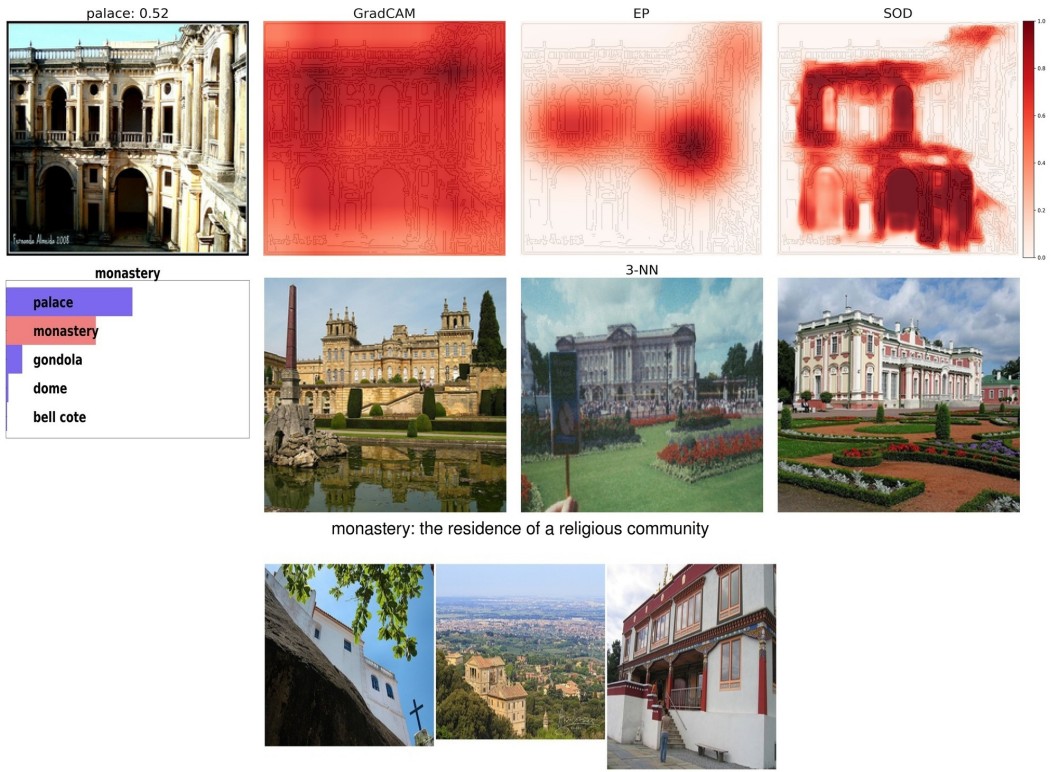

Figure A18: Medium ImageNet image with confusing objects. 3-NN failed to show the difference between "palace" and "monastery" to users.

### A14.3 Easy, AI-misclassified, ImageNet images that were correctly rejected by 3-NN users but not GradCAM and SOD users

3-NN helped users distinguish the two classes by showing contrastive examples (e.g. "walking stick" vs. "african chameleon" in Fig. A19 or "horse cart" vs. "grocery store" in Fig. A20).

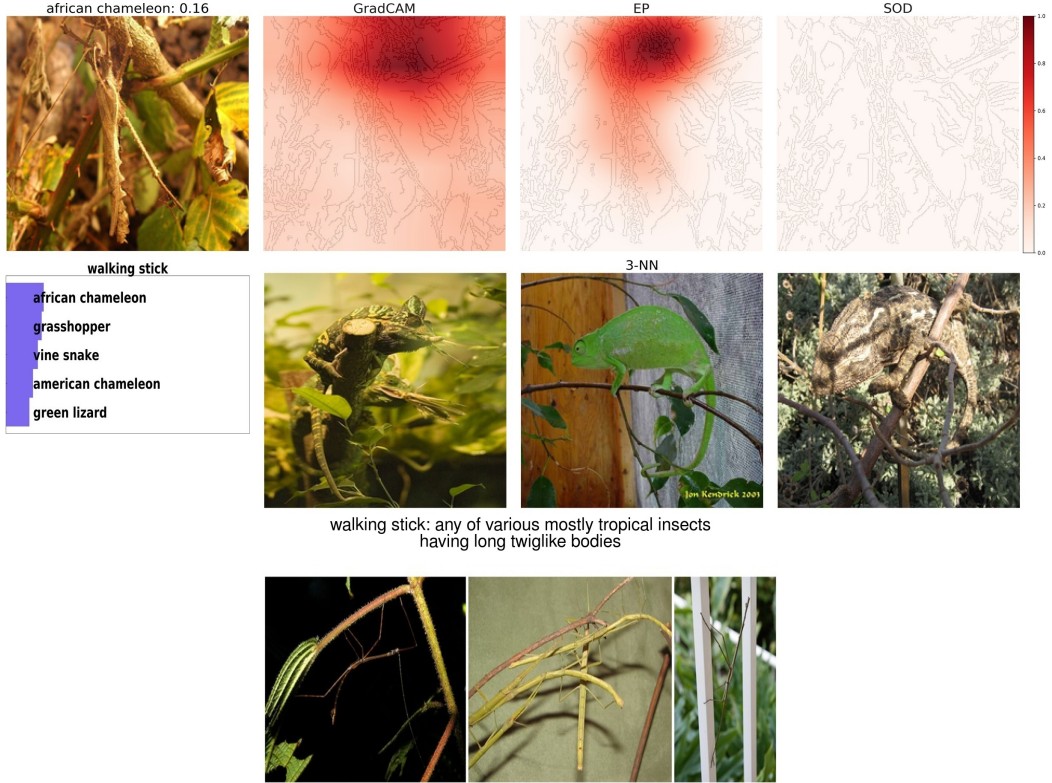

Figure A19: Easy ImageNet image which was clearly misclassified by the AI. 3-NN easily pointed out the difference between "african chameleon" and "walking stick" to users.

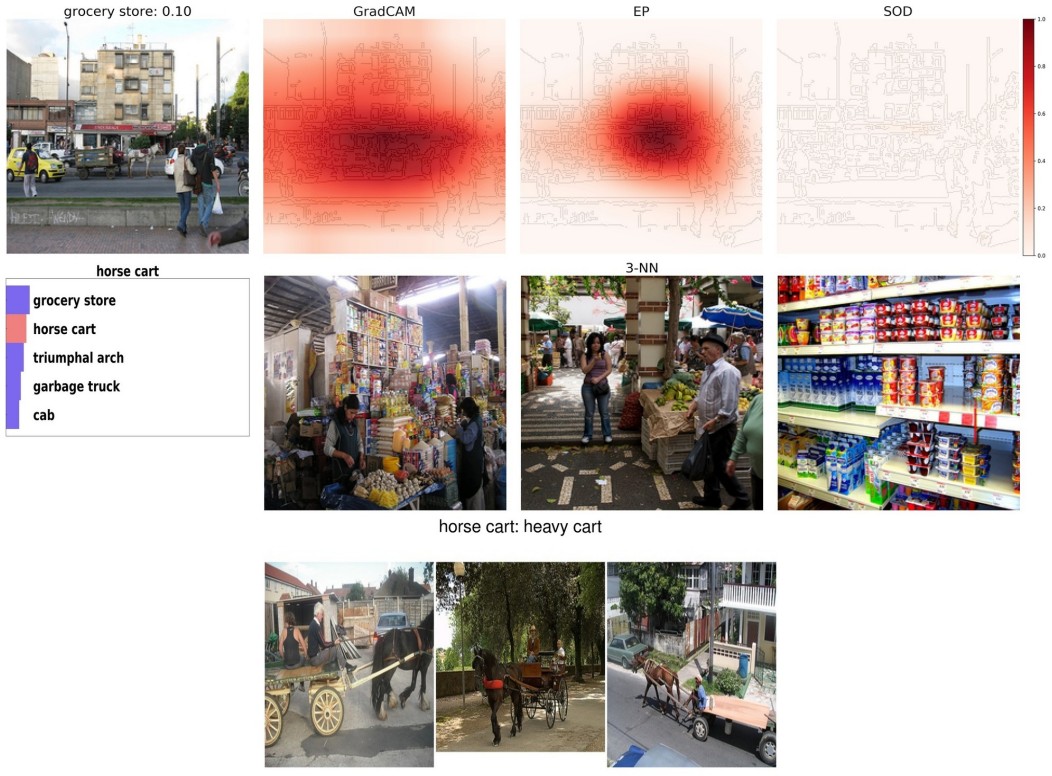

Figure A20: Easy ImageNet image which was clearly misclassified by the AI. 3-NN easily pointed out the difference between "grocery store" and "horse cart" to users.

### A14.4 Adversarial ImageNet images that were correctly rejected by 3-NN users but not GradCAM, EP, or SOD users

**In Fig. 2, what made 3-NN more effective than attribution maps on Adversarial ImageNet?**
As the adversarial attacks fooled AI by small perturbations, the misclassified labels are not far from the ground truth (e.g. bee eater to lorikeet in Fig. A22 or collie to shetland sheepdog in Fig. A26). The highlights of heatmaps focused on parts of the main object and made the explanations compelling to users. 3-NN helped users differentiate the two categories by looking at the constrastive images of the predicted label and the ground truth (Fig. A21 and Fig. A22).

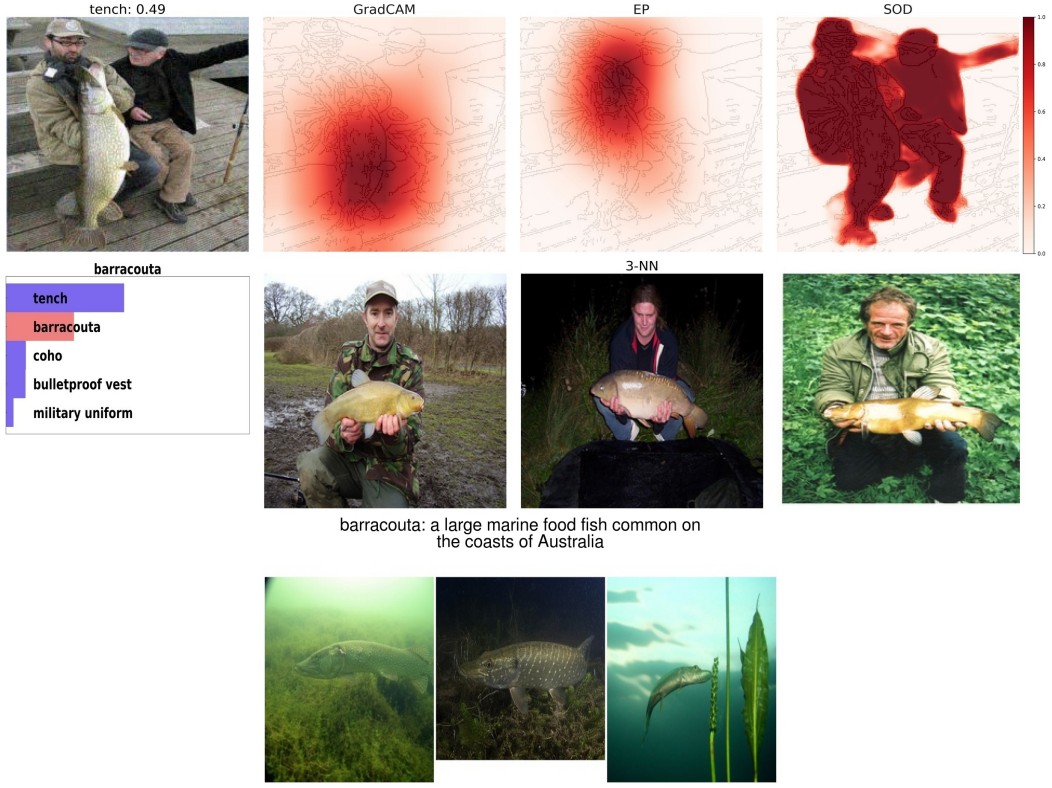

Figure A21: Adversarial ImageNet image of "barracouta" which was misclassified to "tench". Users may use the difference in skin patterns of "tench" and "barracouta" to make decision.

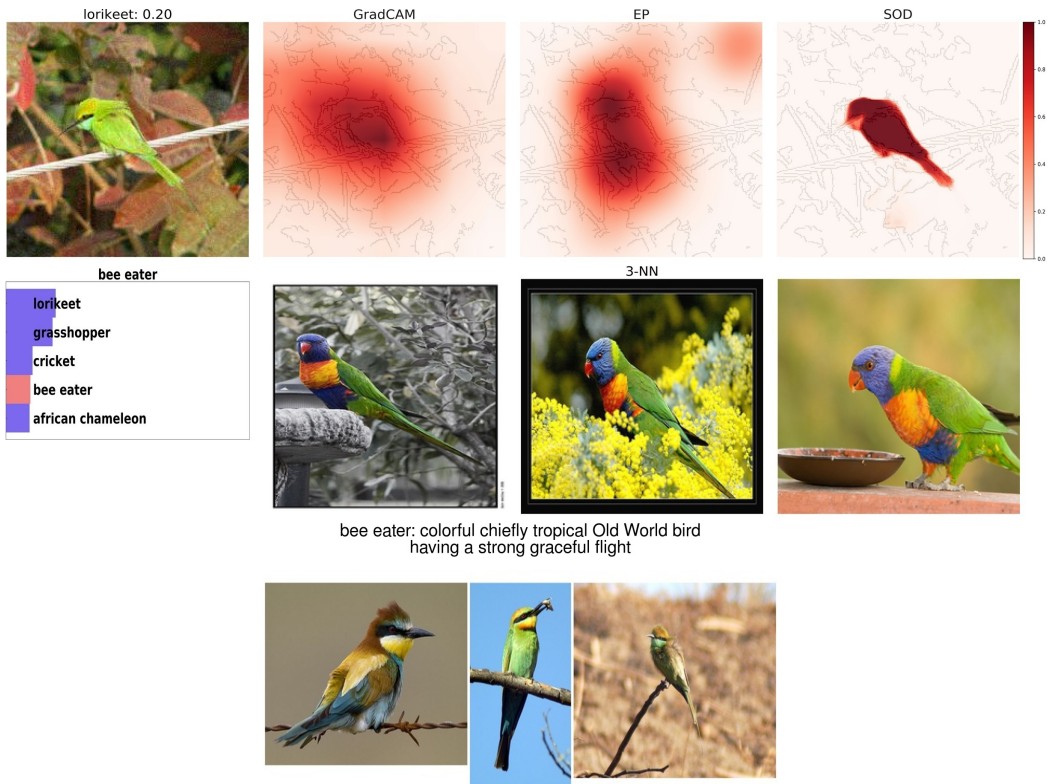

Figure A22: Adversarial ImageNet image of "bee eater" which was misclassified to "lorikeet". 3-NN contrasted these two bird breeds strongly.

### A14.5 AI-misclassified Dogs images that are correctly rejected by SOD users but not GradCAM, EP, or 3-NN users

**In Table A1, what made SOD significantly more effective than other methods on correcting AI-misclassified images of Dogs?** We found that GradCAM and EP often highlighted the entire face of the dogs, which made the heatmaps persuasive to users although the predictions were wrong. Regarding 3-NN, the misclassified category is visually similar to the ground truth (i.e. eskimo dog vs. malamute), which was challenging for lay users to distinguish. We assume that users expected explanations to be as specific and relevant as possible because the differences among breeds are minimal. SOD highlighted the entire body of the dogs (Fig. A23) or even irrelevant areas (Fig. A24). This explains why users with SOD tended to ignore rather than trust the AI. While 3-NN users leveraged the information of nearest neighbors to identify AI's errors, SOD users rejected predictions because of the heatmaps' low quality, which unintentionally improved the accuracy on wrong Dogs images. The rejection rates of SOD users were highest in both ImageNet (38.03%) and Dogs (37.34%) as shown in Table A9. Indeed, due to the high rejection rate of SOD users, the accuracy on correct Dogs images was lowest as shown in Table A1.

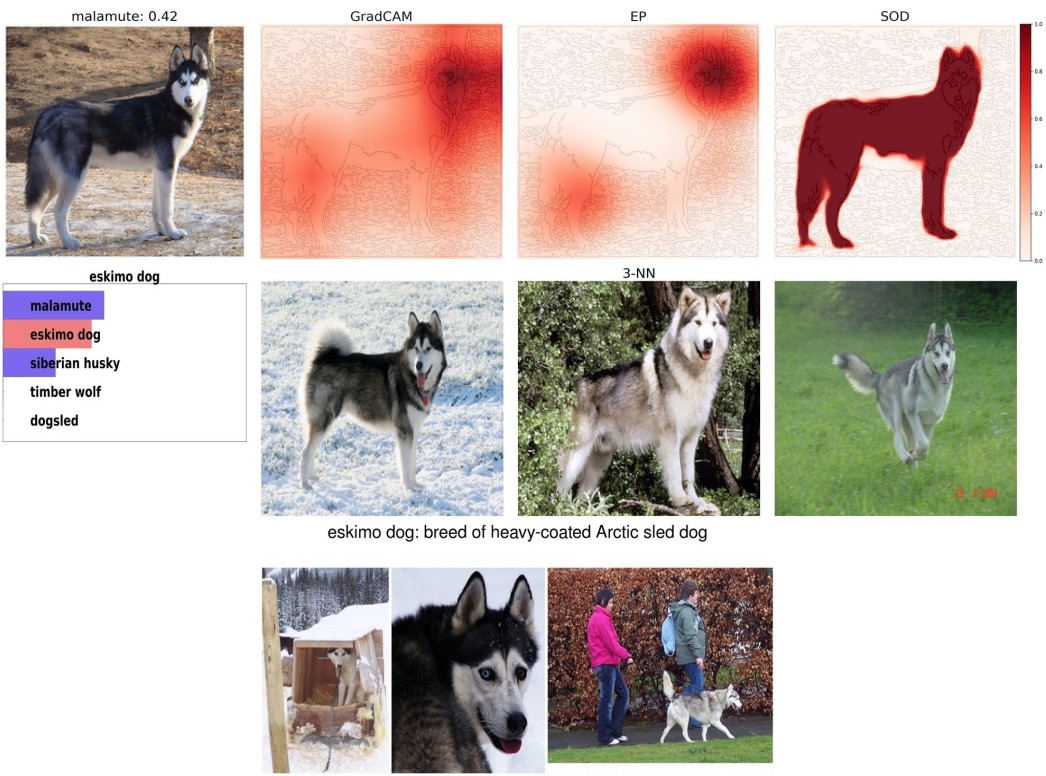

Figure A23: Wrong Dogs image of "eskimo dog" which was misclassified to "malamute". Grad-CAM and EP often highlighted the entire face of the dogs, which makes the heatmaps persuasive to users although the predictions are wrong. For 3-NN, the mislabeled category is visually similar to the ground truth (e.g. eskimo dog vs. malamute), which is challenging for users to distinguish, then they inclined to accept the predictions. SOD always highlights the entire body of the dogs, explaining why users with SOD tended to ignore rather than trust the AI.

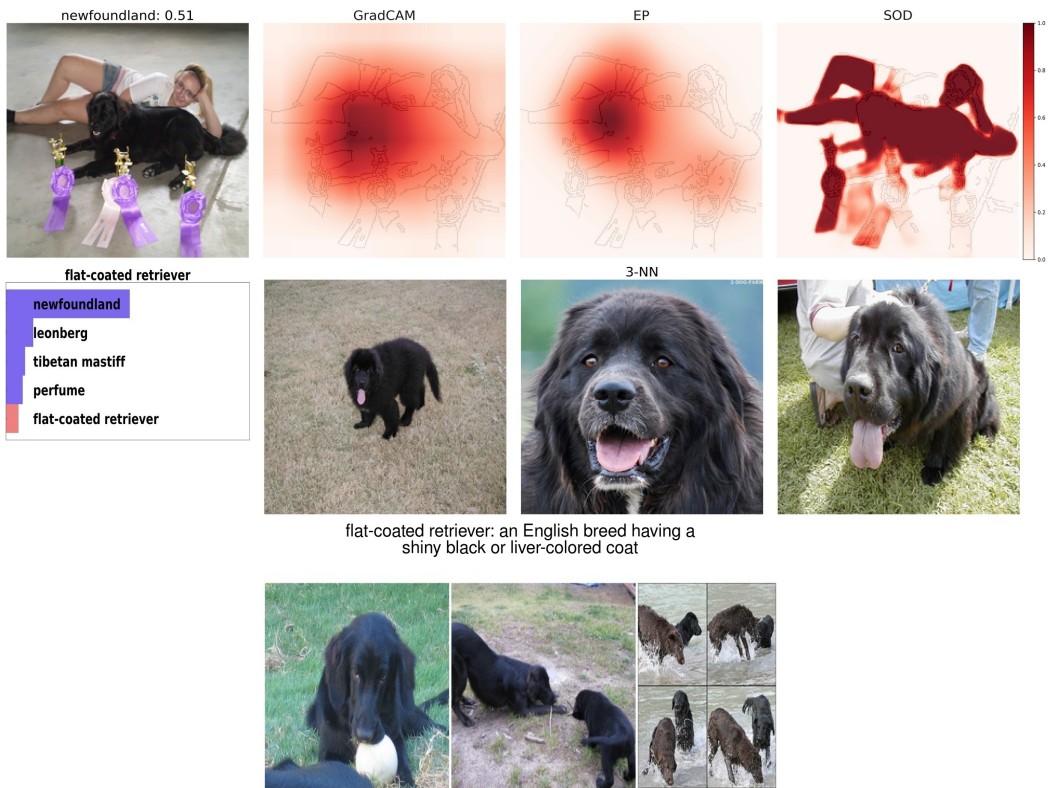

Figure A24: Wrong Dogs image of "flat-coated retriever" which was misclassified to "newfoundland". SOD highlighted irrelevant areas, so users with SOD were more likely to reject.

### A14.6 Adversarial Stanford Dogs images that were correctly rejected by Confidence-only users but not GradCAM, EP, and 3-NN users

**In Fig. 2, why did visual explanations hurt human-AI team performance Adversarial Dogs (the hardest task)?** We found no explanation methods that benefit participants in this task. While GradCAM and EP mostly concentrated on a body part of dogs, 3-NN showed images of an almost identical breed (Fig. A25 and A26). Again, the improvement of SOD came from the bad quality of its heatmap.

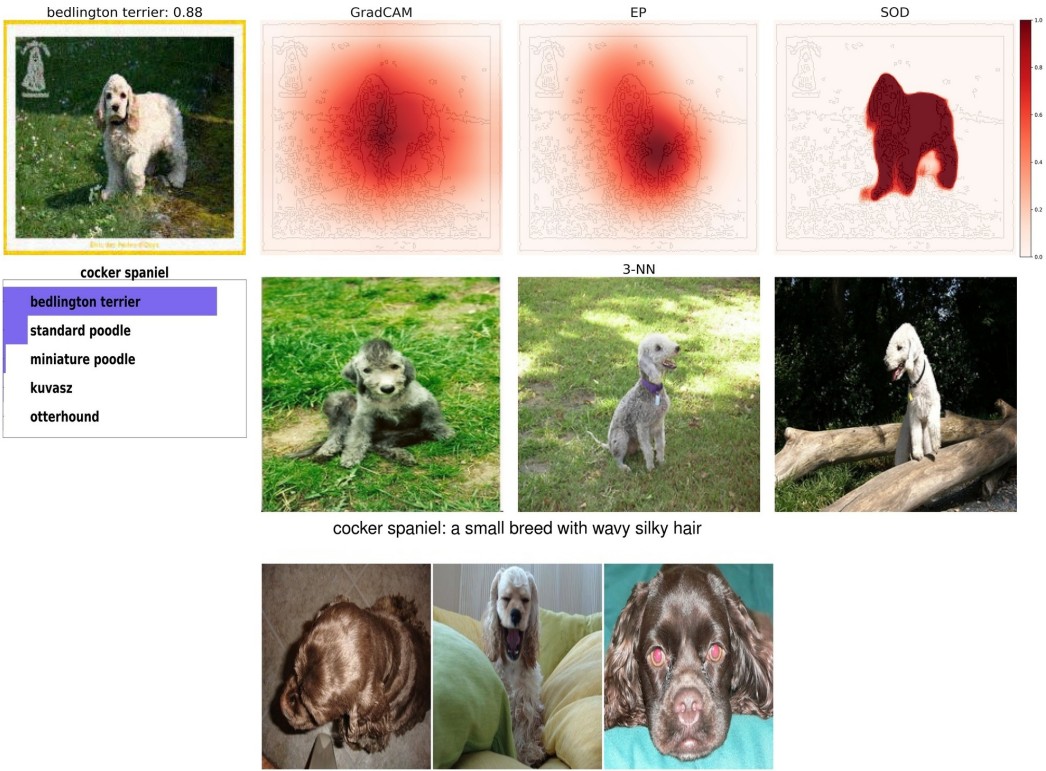

Figure A25: Adversarial Dogs image of "cocker spaniel" which was misclassified to "bedlington terrier". GradCAM and EP concentrated on the belly of dogs, and 3-NN showed images of an almost identical breed, making users trust the prediction.

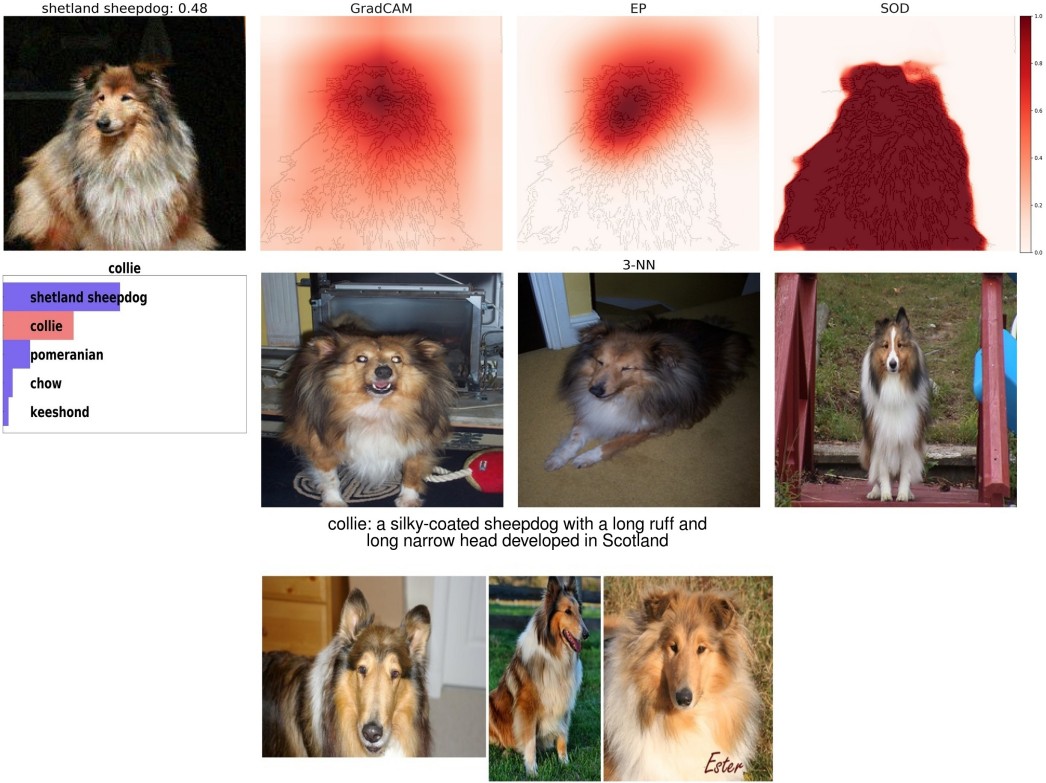

Figure A26: Adversarial Dogs image of "collie" which was misclassified to "shetland sheepdog". GradCAM and EP concentrated on the face of dogs, and 3-NN showed images of a very similar dog breed, making users trust the prediction.