# OpenReview forum: "The effectiveness of feature attribution methods and its correlation with automatic evaluation scores"
_NeurIPS.cc/2021/Conference — NeurIPS 2021 Poster_

### Official Review · Reviewer_62dY · 2021-07-01

**Rating:** 6
**Confidence:** 4

**Summary:**

The paper presents results of a user study that was designed to study the effectiveness of feature attribution methods in assisting human decision makers. Experiments are conducted on Stanford dog dataset and imageNet dataset. The authors mention that feature attribution methods are not any better than 3 nearest neighbor, and that especially for fine grained classification task ( on Stanford dog dataset), feature attribution map in fact hurts.

**Limitations And Societal Impact:**

Please refer to the above response for details pertaining to limitations.

Suggestions for improvement:

1. It will be useful to include a detailed description of the user study, including the rationale for the choice of the specific participants groups and their sizes, participants backgrounds, etc.

2. It is important to compare with other relevant evaluation metrics such as those listed above.

3.  It might be better to consider a high stakes application such as in healthcare, climate, etc. to make the results more compelling.

4. The authors consider a few saliency based methods , primarily GRADCAM. To appeal to the broader research community, it would help to consider a few other models as well.

5. The authors might want to consider some qualitative analysis of feature attribution methods such as those listed below and explain what complementary insights a user study such as their study provides.

a. Epistemic values in feature importance methods: Lessons from feminist epistemology, FAccT 2021
b. If saliency cropping is the answer, what is the question?, CVPR Beyond Fairness Workshop 2021

**Main Review:**

Strengths:

1. The paper studies an interesting problem, to analyze the effectiveness of feature attribution maps in assisting human decision makers. Such studies are necessary given the prevalence of AI models across a variety of high stake applications.

2. The paper is fairly well presented and organized.

3. The authors also list some limitations of their work, giving the reader a better view of the problem scope.

4. Some interesting results are also provided--that feature attribution maps actually hurt in fine grained classification on the dog dataset, and that it is not any better than 3NN in some of the experimental settings.


The following are some issues of concern:

1. Evaluation metrics:

The authors  consider pointing game, intersection over union, and weakly supervised localization for evaluation. However, the following recently proposed metrics related to concept activation vectors, common feature, commonality, etc. are very relevant, especially given that large scale user studies have been performed to learn about their effectiveness. Thus, the metrics used by the authors in evaluation are limited, and the results contained are not necessarily significant.

Benchmarking Attribution Methods with Relative Feature Importance, Yang and Kim, 2019
https://arxiv.org/pdf/1907.09701.pdf

Interpretability Beyond Feature Attribution: Quantitative Testing with Concept Activation Vectors (TCA) , Kim et. al. http://proceedings.mlr.press/v80/kim18d/kim18d.pdf

2. Methodology

a. The authors perform the user study on 320 "lay people" and 11 "experts". This distribution is highly skewed, and it does not therefore represent a reasonable perception across people from these two groups. What was the rationale behind choosing a few "experts" vs so many "lay' people?

b. Sufficient details about these 320 lay people is missing. What were their educational backgrounds, where were they from, etc. These factors can significantly affect how the participants perceive the attribution maps. If these factors were taken into account, then it is not clear how it was done.

c. In the process of giving feedback to the participants, how was it ensured that biases associated with the AI results were not reinforced? In other words, how was it ensured that the feedback did not prime the participants to respond in a certain way? Otherwise, the results are not necessarily accurate.

3. Significance:

The paper starts on a strong note, with reference to high stake applications, and the need for understanding the effectiveness of attribution maps in assisting humans.  However, the authors conduct experiments on ImageNet, and Stanford Dog datasets. It would have been much compelling to show results on a more critical application such as healthcare, environment modeling, etc.  This would enhance the significance of the results.

4. Originality:

In its current form, the novelty of the work seems incremental. The metrics considered are not based on prior art, and also there have been extensive user studies related to understanding saliency maps such as those of [8, 55] that the authors mention.

**Time Spent Reviewing:**

5

---

> ### Author Response · Authors · 2021-08-10
> **Response to reviewer 62dy**
>
> Thank you very much for your positive review and interesting questions!
> We would like to clarify the novelty and significance of our work below.
>
> ### > The following recently proposed metrics (BAM by Yang & Kim 2019; TCAV by Kim et al. 2018) related to concept activation vectors, common feature, commonality, etc. are very relevant, especially given that large scale user studies have been performed to learn about their effectiveness. Thus, the metrics used by the authors in evaluation are limited, and the results contained are not necessarily significant.
>
> We do not understand your criticism. We wish to clarify that the significance and novelty of our work are in a thorough evaluation of the utility of attribution maps in **assisting users** on **image classification**. Previous works have shown that *feature attribution* maps did help humans make more accurate predictions on downstream classification tasks, e.g. on predicting book categories from text [a], movie-review sentiment analysis [a] [e], predicting hypoxemia-risk from medical tabular data [b], or detecting fake text reviews [c] [d]. We are the first to assess _visual_ attribution maps on the popular ImageNet and Stanford Dogs image classification. This utility quantity is NOT measured in the BAM, TCAV paper, or any existing automatic evaluation metrics.
>
> - [a] Schmidt and Biessmann. 2019. Quantifying Interpretability and Trust in Machine Learning Systems. arXiv:1901.08558
> - [b] Lundberg et al. 2018. Explainable machine-learning predictions for the prevention of hypoxemia during surgery. Nature Biomedical Engineering.
> - [c] Lai et al. 2020. "Why is ’Chicago’ Deceptive?" Towards Building Model-Driven Tutorials for Humans. In Proceedings of the 2020 CHI Conference on Human Factors in Computing Systems
> - [d] Lai and Tan. 2019. On Human Predictions with Explanations and Predictions of Machine Learning Models: A Case Study on Deception Detection. FAT 2019.
> - [e] Bansal et al. "Does the whole exceed its parts? the effect of ai explanations on complementary team performance." Proceedings of the 2021 CHI Conference on Human Factors in Computing Systems. 2021.
>
> Could you please elaborate more on which user studies have been performed to study the utility of TCAV or BAM? We are not able to find them.
> BAM is an arXiv publication while the human study in TCAV does not measure human image classification accuracy.
>
>
>
> ### > The authors perform the user study on 320 "laypeople" and 11 "experts". This distribution is highly skewed, and it does not, therefore, represent a reasonable perception across people from these two groups. What was the rationale behind choosing a few "experts" vs so many "lay' people?
>
> Yes, this skewed distribution is common in most other human studies because experts are scarce, and recruiting them to perform human studies is prohibitively expensive.
> TCAV [1] has only 1 medical expert, ChexNet [2] has 4 radiologists, and [3] has 10 experts in their respective experiments. Most papers only evaluate their interpretability methods on laypeople and only a few are tested on both experts and laypeople (please see Table 4 of [3] https://openreview.net/pdf?id=QO9-y8also- ).
> Despite the different sizes, **both laypeople and expert studies consistently found 3-NN to outperform GradCAM** (statistical significance found under Mann Whitney U-test).
>
> - [1] Kim et al. 2018. Interpretability beyond feature attribution: Quantitative testing with concept activation vectors. ICML.
> - [2] Rajpurkar et al. (2017). Chexnet: Radiologist-level pneumonia detection on chest x-rays with deep learning. arXiv preprint arXiv:1711.05225.
> - [3] Borowski et al. (2021). Exemplary Natural Images Explain CNN Activations Better than Feature Visualizations. ICLR 2021.
>
>
> ### > Details about these 320 laypeople are missing. What were their educational backgrounds, where were they from, etc.
>
> Thank you for your suggestion!
> We did consider our user pool carefully and will include the details in Appendix (previously removed from main text due to space).
>
> As written in L128, we only use a single criterion for filtering users: Users must be native English speakers. We think this is required for our study (and any study that uses English) since the training, instructions, and ImageNet labels are in English. We used this filter to avoid the cases where users make arbitrary decisions without understanding some words.
> Prolific shows that our users are diverse, aging from 18-77 (median=31) and coming from a diverse set of countries (US, UK, Poland, India, Korea, Canada, Australia, South Africa, etc.). Please see Prolific for more description of their online userbase, which, according to a study, is more reliable than AMT Turkers [a].
>
> - [a] Peer et al. 2017. Beyond the turk: Alternative platforms for crowdsourcing behavioral research. Journal of Experimental Social Psychology
>
> ### > The metrics considered are not based on prior art, and also there have been extensive user studies related to understanding saliency maps such as those of [8, 55] that the authors mention.
>
> We argue that one important utility of XAI methods is to assist users in decision-making in downstream tasks. W.r.t. to the prior art, we are **the first** to study the utility of attribution methods on ImageNet classification and Stanford Dog fine-grained image classification.
> This is important given that ImageNet has been extensively used for demonstrations in hundreds of feature-attribution papers.
>
> We do not understand the criticism that _"The metrics considered are not based on prior art"_. Pointing Game, IoU or WSL are not recent methods, but they are being used heavily in the literature to assess attribution maps (see L195-L206). Please note that recent metrics e.g., (BAM; Yang and Kim 2019) or (Adebayo et al. Debugging Tests for Model Explanations. NeurIPS 2020) evaluate attribution maps on **synthetic data**. Here, we test users on the **real** dataset of ImageNet and Stanford Dogs.
>
> ### > The authors consider a few saliency-based methods, primarily GradCAM. To appeal to the broader research community, it would help to consider a few other models as well.
>
> We had also tested with other attribution methods (e.g. VanillaGradient [7], IntegratedGradient [8], SHAP[9]) but excluded them from the study due to their noisy visualizations (which we found to have very little utility, same as in [10]). Therefore, we only chose GradCAM, EP, and SOD in our final experiments. We clarify this in our camera-ready paper. Thank you!
>
> - [7] Simonyan et al.. Deep inside convolutional networks: Visualising image classification models and saliency maps. In ICML workshop 2014.
> - [8] Sundararajan et al. Axiomatic attribution for deep networks. ICML 2017.
> - [9] Lundberg et al. A unified approach to interpreting model predictions. NeurIPS 2017.
> - [10] Adebayo et al. Debugging Tests for Model Explanations. NeurIPS 2020.
>
>
> ### > The authors might want to consider some qualitative analysis of feature attribution methods such as those [a,b] listed below and explain what complementary insights a user study such as their study provides.
> **a. Epistemic values in feature importance methods: Lessons from feminist epistemology, FAccT 2021 b. If saliency cropping is the answer, what is the question?, CVPR Beyond Fairness Workshop 2021.**
>
> Thank you for pointing us to these two interesting papers! However, we do not understand which quantity of feature attribution you are suggesting we study. We'd love to know clearer suggestions if you have.

---

> > ### Comment · Reviewer_62dY · 2021-08-29
> > **Thank you for the response**
> >
> > Thanks for the clarifications. I think evaluating the efficacy of the method on real world applications (such as a medical dataset) would be very beneficial. That said, I think the authors have provided a ground work for such further studies which can be helpful in enhancing AI ethics. I therefore increase my score.

---

### Official Review · Reviewer_Fe2L · 2021-07-12

**Rating:** 6
**Confidence:** 5

**Summary:**

The paper investigates if current widely used evaluation metrics for saliency maps or explainable visual AI like pointing game, weakly supervised localization are trustable by conducting a very extensive user study. Some interesting conclusions and observations are given. In this sense, this is a good paper.

**Limitations And Societal Impact:**

yes

**Main Review:**

The paper claims it is the ﬁrst, large-scale user study, but 320 users seem to be not large enough when comparing with previous papers. It is just a total submission. In line 133, each experiment was conducted by 30 users. In the paper (Teaching Categories to Human Learners with Visual Explanations), even 40 users were recruited for each setting. So I think the paper should tone down the claim. The same for some conclusions, because only dogs data is used, this can not generalize to all fine-grained domains.

Also, the training step is a machine teaching task, how to select the samples and how to decide their order shown to the user. This is critical, there are many machine teaching papers, for example
1)Teaching Categories to Human Learners with Visual Explanations
2)Near-Optimally Teaching the Crowd to Classify
3)Interpretable Machine Teaching via Feature Feedback
This has an impact on the quality of the user training. The suboptimal teaching samples may cause the observation less trustable.

I'm curious about the variance of the user study results. Some statistic analysis should be included.

Because English is used to filter users. Not sure if this makes the conclusion have a bias on the race.


**Time Spent Reviewing:**

4

---

> ### Author Response · Authors · 2021-08-10
> **Response to reviewer Fe2L**
>
> Thank you for your positive comments and valuable feedback! Please kindly find our replies below.
>
> ### > The paper should tone down the claim. The same for some conclusions, because only dogs data is used, this can not generalize to all fine-grained domains.
> We agree and will revise the claims and conclusions in the paper to avoid unintentionally causing readers' misinterpretation of our results' generalization and novelty (e.g. L33).
> That is, we will add proper identifiers to communicate that we are the first to study the utility of attribution maps in **assisting humans in image classification on ImageNet and Stanford Dogs**.
>
> ### > Also, the training step is a machine teaching task, how to select the samples and how to decide their order shown to the user. This is critical, there are many machine teaching papers.
> Thank you for making this great point!
> Yes, we will add to the **Limitations** section that our study does not consider state-of-the-art methods for training humans as in the mentioned papers. We will cite and discuss these three papers.
> Training humans is a topic that currently most XAI studies do not pay enough attention to partly because each explanation method may have a different optimal training procedure.
>
> ### > I'm curious about the variance of the user study results. Some statistic analysis should be included.
>
> Yes, we revise the paper to show standard deviations in Fig. 2 (as done in Table 1 for expert users) and findings with statistical-significance test results.
>
> **ImageNet:** On our controlled image sets (correctly-labeled, incorrectly-labeled, and adversarial images; see Sec. 2.4.3), the user-accuracy scores of each method have the following statistics:
>
> | Method   |      Mean      |  Std |
> |----------|:-------------:|:------|
> | Confidence (no visual explanation) | 72.44 	 | 8.25 |
> | GradCAM |	 72.58 |8.11 |
> | EP 	| 73.85 	 | 6.88 |
> | SOD | 	 72.06 	 | 7.63 |
> | 3-NN |	 **76.08** 	 | **5.86** |
>
> - We found the findings that _3-NN is better than SOD_ and _3-NN is better than GradCAM_ to be *statistically significant* (Mann Whitney U-test; p < 0.035). In the expert study (see Table 1), we also found that _3-NN is better than GradCAM_ (both in terms of means and std).
> - The differences among feature attribution methods are NOT statistically significant.
> - To help readers understand the utility of these explanation methods on the _uncontrolled_ ImageNet distribution, we also converted the accuracy on the controlled bins into the accuracy on the natural ImageNet distribution (Fig. A1 in Appendix). On random ImageNet images, 3-NN mean accuracy remains higher than that of feature attribution methods (Fig. 2).
>
> **Stanford Dogs:** On our controlled image sets (correctly-labeled, incorrectly-labeled, and adversarial images; see Sec. 2.4.3), the user-accuracy scores of each method have the following statistics:
>
> | Method   |      Mean      |  Std |
> |----------|:-------------:|:------|
> | Confidence (no visual explanation) | **61.71** | 11.39 |
> | GradCAM |	 60.56 | 9.27 |
> | EP 	| 56.67 | 10.57 |
> | SOD | **61.67** | 10.87 |
> | 3-NN | 57.20 | 10.58 |
>
> We found 3-NN and EP to be statistically significantly lower than the Confidence and SOD baselines (Mann Whitney U-test; p < 0.024). We found no significant differences among other pairs of methods.
> That is, interestingly, on fine-grained Stanford Dog classification, visual explanations hurt the user accuracy on image classification.
> This negative result serves as a good starting point for inventing AIs that assist humans' decision-making better in the future.
>
> ### > Because English is used to filter users. Not sure if this makes the conclusion have a bias on the race.
> Thank you for pointing this out! We will include more user pool demographics data in the paper.
>
> As written in L128, we only use a single criterion for filtering users: Users must be native English speakers. We think this is required for our study (and any study that uses English) since the training, instructions, and ImageNet labels are in English. We used this filter to minimize the cases where users make arbitrary decisions without understanding some words. Prolific shows that our users are diverse, aging from 18-77 (median=31) and coming from a diverse set of countries (US, UK, Poland, India, Korea, Canada, Australia, South Africa, etc.).

---

> > ### Comment · Reviewer_Fe2L · 2021-08-30
> > **post rebuttal**
> >
> > Thanks for the effort of the authors. After reading the response, most of my concerns have been solved. I'd like to maintain my original positive score.

---

### Official Review · Reviewer_wdnC · 2021-07-13

**Rating:** 7
**Confidence:** 4

**Summary:**

The main contribution of this paper is a thorough and well-designed user study towards using feature attribution maps, i.e. GradCAM, in practice to help human users understand prediction results of neural networks. Following the user study, the author finds several existing metrics in evaluating the goodness of feature attributions do not align with the user’s gains in having feature attributions and conclude that in the evaluated scenarios features attributions are not more useful than K nearest neighbor explanations.

**Limitations And Societal Impact:**

 The authors adequately addressed the limitations and potential negative societal impact of their work.

**Main Review:**

In general the paper has presented very interesting results and conducted a well-designed user-study that has a larger group of interviewees than previous ones. The presentation is easy to follow. My major concern is the fundamental assumptions about attribution methods made by the authors. I will elaborate more in the *Quality* paragraph.

### Originality
A related work from Adebayo et. al. [1] that leverages user study to evaluate the feature attribution methods is missing and may potentially be very important to compare with the paper’s findings. Though Adebayo et. al. [1] has less interviewees compared to this study but it contains more tests for a different group of feature attributions that this paper does not consider as *state-of-the-art*. Can the authors elaborate more about the important difference between this paper and Adebayo et. al. [1] that makes the user study present in this paper as the *“the first, large-scale user study to shed light on the effectiveness of AMs”* as claimed in the paper?

### Quality
This paper has presented a well-designed user-study pipeline to evaluate the performance of explanations methods for the downstream users. I have no concerns on the dataset distribution, network architectures, and the details about the user-study. My only concern is that there is not enough justification why the way the feature attribution is used in this paper agrees with what feature attributions are designed for in the first place.

1. Firstly, this paper expects users to leverage feature attribution in identifying whether the prediction matches the groundtruth. However, feature attributions, as its name suggests, is only a way to attribute the output scores over the input (or internal) features. The target question an attribution answers is more like **“how features contribute to the quantity of interest, i.e. the prediction”** instead of **“is the prediction accurate”** [2]. Justifications on why feature attributions are expected to be the one, among other explanations tools,  that should answer the question this paper designs are currently missing. By other explanations tools, an example is instance-based method [3, 4].

2. Secondly, the paper concludes that attributions with high localization accuracy do not help humans to make the correct decisions (line 297-299). Because the localization metrics discussed in this paper are based on the bounding boxes generated by humans, these metrics are actually evaluating whether attributions match the human’s prior. Therefore, the authors need to explain why the optimal attribution method that perfectly matches the human’s prior is supposed to help a human who already knows how to draw the bounding box in general. I agree that evaluation metrics need improvement but the current way of concluding the results does not help us to understand why the localization metrics do not calibrate with the user-study. More analysis will help the reader to gain insights from the paper’s result.

### Clarity
The authors have documented their process in a reproductive way. The paper is well-organized and well-written.

### Significance
This work is very significant for showing whether explanations can actually help humans in practice to understand the predictions of deep classifiers.

[1] Adebayo, J., Muelly, M., Liccardi, I., & Kim, B. (2020). Debugging Tests for Model Explanations. NeurIPS 2020.

[2]K. Leino, S. Sen, A. Datta, M. Fredrikson and L. Li, "Influence-Directed Explanations for Deep Convolutional Networks," 2018 IEEE International Test Conference (ITC), 2018.

[3] Yeh, C., Kim, J.S., Yen, I.E., & Ravikumar, P. (2018). Representer Point Selection for Explaining Deep Neural Networks. NeurIPS 2018.

[4] Koh, P. W., & Liang, P. (2017, July). Understanding black-box predictions via influence functions. In ICML 2017.

**Time Spent Reviewing:**

7

---

> ### Author Response · Authors · 2021-08-10
> **Response to reviewer wdnC**
>
> Thank you for your very positive feedback and very insightful questions!
> Please kindly find our replies below.
>
>
> ### > A related work from Adebayo et. al. NeurIPS 2020 that leverages user study to evaluate the feature attribution methods is missing and may potentially be very important to compare with the paper’s findings. Can the authors elaborate more about the important difference between this paper and Adebayo et. al. [1] that makes the user study present in this paper as “the first, large-scale user study to shed light on the effectiveness of AMs” as claimed in the paper?
>
> Thank you! We will discuss Adebayo et al. NeurIPS 2020 in our camera-ready version.
>
> - We wish to clarify our _identifier_ here: We are the first to study the utility of AMs **in assisting humans in ImageNet and Stanford Dog classification**.
> However, there are other purposes of AMs that we do not study in this paper e.g. model debugging as in Adebayo et al. NeurIPS 2020 or teaching humans. We will make this point clearer in the paper to avoid misinterpretation about the novelty.
> - Adebayo et al. 2020 presented a very interesting study on synthetic data and found that humans mostly rely on predicted labels rather than AMs to debug a model. Aligned with their work, our study on real images did not find any statistically significant differences between Confidence (i.e. no visual explanations) and three feature attribution methods (EP, GradCAM, SOD) **on ImageNet**. Furthermore, on **Stanford Dogs**, users without explanations (Confidence) even outperform users with 3-NN and EP heatmaps (Mann Whitney U-test p-value < 0.024).
>
>
> ### > Justifications on why feature attributions are expected to be the one, among other explanations tools, that should be useful to humans in image classification?
>
> That's a great question!
> One of the desired capabilities of explainable/interpretable AI systems is that they **assist humans** on the downstream task, here **image classification**. That is, humans may benefit from working with a companion AI. Previous works have shown that **feature attribution** maps did help humans make more accurate predictions on downstream classification tasks, e.g. on predicting book categories from text [a], movie-review sentiment analysis [a] [e], predicting hypoxemia-risk from medical tabular data [b], or detecting fake text reviews [c] [d]. This success of explanations in improving human performance in non-image domains motivates us to test the same hypothesis on **image** classification. As in [d], we consider both heatmap-based and example-based methods. [f] also tested the effectiveness of attribution maps in helping users predict age from facial photos.
>
> When an attribution map highlights the non-relevant input region, ideally human users would choose to not use AI predictions but instead will make their own decision. When an attribution map highlights relevant regions, users may focus their attention on such areas.
> We agree that the currently tested AMs are not yet optimized for improving human accuracy on image classification. However, our study serves as an important initial finding for future research in this human-AI collaboration direction.
>
> We will make this point clearer in the camera-ready version. Thank you!
>
>
> - [a] Schmidt and Biessmann. 2019. Quantifying Interpretability and Trust in Machine Learning Systems. arXiv:1901.08558
> - [b] Lundberg et al. 2018. Explainable machine-learning predictions for the prevention of hypoxemia during surgery. Nature Biomedical Engineering.
> - [c] Lai et al. 2020. "Why is ’Chicago’ Deceptive?" Towards Building Model-Driven Tutorials for Humans. In Proceedings of the 2020 CHI Conference on Human Factors in Computing Systems
> - [d] Lai and Tan. 2019. On Human Predictions with Explanations and Predictions of Machine Learning Models: A Case Study on Deception Detection. FAT 2019.
> - [e] Bansal et al. "Does the whole exceed its parts? the effect of ai explanations on complementary team performance." Proceedings of the 2021 CHI Conference on Human Factors in Computing Systems. 2021.
> - [f] Chu et al. 2020. Are visual explanations useful? a case study in model-in-the-loop prediction. arXiv preprint arXiv:2007.12248.
>
>
> ### > The authors need to explain why the optimal attribution method that perfectly matches the human’s prior is supposed to help a human who already knows how to draw the bounding box in general.
>
> Thank you for a great question! Please find our answer below, which we will include in the final paper.
>
> In comparison of attribution methods, a variety of works used localization accuracy to claim the quality of attribution maps (e.g. CAM [a], Table 1 in GradCAM [b], Table 3 in GradCAM++ [c], Table 2 in XRAI [d], or Table[1] in Integrated-GradCAM[e]). But it is unclear whether such methods with high localization abilities can help humans in making better decisions. Our research is motivated by two facts: (1) AIs are performing very well on i.i.d. test sets; and (2) existing attribution methods are able to show humans what AIs are focusing on to make decisions. Therefore, the research question is: Would humans make more accurate decisisions if they know the predictions of a high-performing AI and its "attention" map for a given input image?
>
> Here, our work attempted to connect the dots between the quality of attribution maps (in localization errors) vs. their utility in assisting humans in image classification.
>
> - [a] Zhou et al. (2016). Learning deep features for discriminative localization. CVPR 2016
> - [b] Selvaraju et al. Grad-cam: Visual explanations from deep networks via gradient-based localization. CVPR 2017
> - [c] Chattopadhay et al. Grad-cam++: Generalized gradient-based visual explanations for deep convolutional networks. WACV.
> - [d] Kapishnikov et al. Xrai: Better attributions through regions. CVPR 2019
> - [e] Sattarzadeh et al.. Integrated Grad-Cam: Sensitivity-Aware Visual Explanation of Deep Convolutional Networks Via Integrated Gradient-Based Scoring. In ICASSP 2021.
>
>
> ### > Why do the localization metrics not calibrate with the user study?
>
> One reason we found is that because a localization region often covers the entire main object and therefore does not inform users well on whether the predicted label is correct or not. Our study showed that feature attribution maps often highlight the main object regardless of whether AI is correct or not (L277).
> Additionally, the poor utility of attribution maps in our work suggests that the interaction between humans and the explanations is a good research direction in the future to study in order for users to make the most out of attribution maps.

---

> > ### Comment · Reviewer_wdnC · 2021-08-29
> > **I appreciate the response and I will increase my score accordingly.**
> >
> > Hi authors,
> >
> > My apology for the late response. After reading the authors' response on my feedback, I decide to increase my score from 5 to 7 because my concerns are well-addressed. I am looking forward to seeing the authors to include the discussion into their future version of the paper if accepted. I believe this paper is valuable to serve as a building block in measuring how explanations can be used in practice to help human more than just finding important features despite of the current limitations presented in the paper and pointed out by other reviewers.

---

### Official Review · Reviewer_Tf4g · 2021-07-16

**Rating:** 7
**Confidence:** 4

**Summary:**

This paper performs a human evaluation study on feature attribution style explanations.  The main conclusions include:

1. Attribution methods (AMs) are not more effective than an "explanation" which consists of three nearest neighbor images.  In some settings (adversarial setting on Stanford Dogs) presenting confidence scores to humans is most helpful.

2.  Current automatic metrics to measure attribution methods don't correspond well with this human evaluation.

3.  When testing on AI-expert users, nearest neighbor explanations are more helpful.



**Ethical Concerns:**

None.

**Limitations And Societal Impact:**

There is a limitations section but no specific societal impact section.  It would be good to include a societal impact statement, but I don't think this work contains any specific ethical concerns.

Though not listed under societal impact, I appreciated the description of how users were paid.  I wonder where participants were located and how the $10.20 compared to minimum wage in that region.

**Main Review:**

Overall, I am positive about this submission.

*Originality*

There have been human evaluations of attribution methods in prior papers, but I think this paper adds a few new things which are valuable:

* evaluation of multiple methods on multiple different datasets
* inclusion of experiments on experts and non-experts
* inclusion of a simple form of explanation -- nearest neighbors
* study comparing automatic metrics to human evaluation

This paper could be more useful if it could provide concrete suggestions on where we go as a field.  In particular:

* after this study do researchers think that our current attribution methods are bad, but there could exist attribution methods that are helpful for this task or do researchers thing that attribution methods in general are weak and we should look for other forms of explanation?
* this paper looks at one kind of study: can humans and AI work together to reject bad decisions?  Are there other ways attribution methods could help humans not covered in this study (something to potentially cover in limitations)


*Quality*

I found this paper easy to read and the methodology appears sound.  However, I had a few comments/questions:

* What would the error bars be in Fig 3?  Are the differences actually statistically significant?  If the results aren't statistically significant (e.g., 3-NN, AM, and no explanation are the same), I don't think this significantly changes the current conclusions (currently AM isn't really any better than 3-NN), but I think it is important to include.
* In the human evaluation, my understanding is that each human saw one setting (e.g., each human saw the 3-NN setting).  Did author's consider showing users multiple settings (and permuting order in which users saw the settings?).  I am wondering if there is some variation btwn humans (e.g., familiarity with dog breeds) that could be controlled for better in this way.
* I really liked the inclusion of the SOD baseline.  Surprisingly, SOD does quite well on natural ImageNet and Adversarial Stanford Dogs.  I am wondering why this might be (part of me wonders if this is because the differenced between all the methods are not statistically significant).
* One concern I have with these sorts of studies is that the UI may not be optimized; in particular for most AM the heat map covers the part of the image important for classification.  Did authors consider this or experiment with this?

Additionally I believe there are some missing citations (other works which have evaluated explanation models):

Chandrasekaran et al.  "Do Explanations make VQA Models more Predictable to a Human?"  EMNLP 2018.
Hase et al.  "Evaluating Explainable AI: Which Algorithmic Explanations Help Users Predict Model Behavior?"  ACL 2020.

*Clarity*

Overall I found this paper clear and easy to read.

*Significance*

Over the last few years, there have been critiques over attribution methods and explanations in general.  This paper provides something new to the conversation (see under originality) so I think it is useful for the community.  I do have some concerns about the way experiments were carried out (mainly significance tests), but I imagine this can be cleared up in the rebuttal.

**Time Spent Reviewing:**

1.5 hours.

---

> ### Author Response · Authors · 2021-08-10
> **Response to reviewer Tf4g**
>
> Thank you for your positive feedback on our work and thoughtful comments! Please find our replies below.
>
> ### > After this study do researchers think that our current attribution methods are bad, but there could exist attribution methods that are helpful for this task or do researchers think that attribution methods, in general, are weak and we should look for other forms of explanation?
>
> Thank you for a great question!
> We envision that an important utility of XAI methods is to assist humans in the actual downstream tasks, here image classification. However, we think that attribution methods, in general, are limited in helping users in image classification because **they only use the evidence from the input alone** to form an explanation. As suggested by the superiority of 3-NN in this study, we think that a more promising XAI approach for human-AI team collaboration is when the XAI system searches for an explanation in an external knowledge-base to inform humans. At the moment, the existing methods in this category are prototype-based and visually-grounded explanations.
>
> ### > This paper looks at one kind of study: can humans and AI work together to reject bad decisions? Are there other ways attribution methods could help humans not covered in this study (something to potentially cover in limitations)
>
> We wish to clarify that we study how humans and AI work together in image classification, which requires humans to correctly accept AI's correct predictions and reject AI's incorrect predictions (see Table A1 in Appendix for the breakdown performance of each explanation method).
> Yes, attribution maps could help humans in other ways that we did not study e.g., model debugging [a] [b], teaching humans (e.g. how to distinguish fine-grained butterfly types) [c].
>
> - [a] Debugging Tests for Model Explanations. NeurIPS 2020.
> - [b] Analyzing Classifiers: Fisher Vectors and Deep Neural Networks. CVPR 2015.
> - [c] Teaching Categories to Human Learners with Visual Explanations. CVPR 2018.
>
>
> ### > What are the conclusions with the consideration of error bars and statistical significance tests?
>
> Yes, we revise the paper to show standard deviations in Fig. 2 (as done in Table 1 for expert users).
>
> **ImageNet:** On our controlled image sets (correctly-labeled, incorrectly-labeled, and adversarial images; see Sec. 2.4.3), the user-accuracy scores of each method have the following statistics:
>
> | Method   |      Mean      |  Std |
> |----------|:-------------:|:------|
> | Confidence (no visual explanation) | 72.44 	 | 8.25 |
> | GradCAM |	 72.58 |8.11 |
> | EP 	| 73.85 	 | 6.88 |
> | SOD | 	 72.06 	 | 7.63 |
> | 3-NN |	 **76.08** 	 | **5.86** |
>
> - We found the conclusion that _3-NN is better than SOD_ and _3-NN is better than GradCAM_ to be *statistically significant* (Mann Whitney U-test; p < 0.035). In the expert study (see Table 1), we also found that _3-NN is better than GradCAM_ (both in terms of means and std).
> - The differences among feature attribution methods are NOT statistically significant.
> - To help readers understand the utility of these explanation methods on the _uncontrolled_ ImageNet distribution, we also converted the accuracy on the controlled bins into the accuracy on the natural ImageNet distribution (Fig. A1 in Appendix). On random ImageNet images, 3-NN mean accuracy remains higher than that of feature attribution methods (Fig. 2).
>
> **Stanford Dogs:** On our controlled image sets (correctly-labeled, incorrectly-labeled, and adversarial images; see Sec. 2.4.3), the user-accuracy scores of each method have the following statistics:
>
> | Method   |      Mean      |  Std |
> |----------|:-------------:|:------|
> | Confidence (no visual explanation) | **61.71** | 11.39 |
> | GradCAM |	 60.56 | 9.27 |
> | EP 	| 56.67 | 10.57 |
> | SOD | **61.67** | 10.87 |
> | 3-NN | 57.20 | 10.58 |
>
> We found 3-NN and EP to be statistically significantly lower than the Confidence and SOD baselines (Mann Whitney U-test; p < 0.024). We found no significant differences among other pairs of methods.
> That is, interestingly, on fine-grained Stanford Dog classification, visual explanations hurt the user accuracy on image classification.
> This negative result serves as a good starting point for inventing AIs that assist humans' decision-making better in the future.
>
> ### > In the human evaluation, my understanding is that each human saw one setting (e.g., each human saw the 3-NN setting). Did author's consider showing users multiple settings (and permuting order in which users saw the settings?).
>
> For each user, we only asked them to use one single explanation method (e.g. 3-NN or EP). We thought that asking a single user to try multiple methods will introduce errors due to context-switching and makes it more challenging to train a single user.
> We do show each user two types of images: natural and adversarial images, and the image orders were randomly chosen.
>
> ### > I am wondering if there is some variation between humans (e.g., familiarity with dog breeds) that could be controlled for better in this way.
> Yes, we had considered controlling for human expertise in dog species. However, this would be a standalone study by itself as determining correctly human expertise level is non-trivial. Alternatively, one could use exclusively real dog experts in an in-person study; however, such study would be small in scale. Here, we chose to study general laypeople with no assumption about their expertise in dog classification. We discussed this point in L349 under **Limitations**.
>
> ### > I really liked the inclusion of the SOD baseline. Surprisingly, SOD does quite well on natural ImageNet and Adversarial Stanford Dogs. I am wondering why this might be (part of me wonders if this is because the differences between all the methods are not statistically significant).
>
> Thank you! On Stanford Dogs, SOD outperforms both 3-NN and EP (statistical significance under Mann Whitney U-test; p < 0.024). An explanation is that, as in L240, SOD users tend to reject more often than other users (Sec. A9 in Appendix), inadvertently causing a high human-AI team accuracy on AI-misclassified images (Sec. A13.5).
> On natural ImageNet, the difference between SOD and every other method is not statistically significant.
>
> ### > One concern I have with these sorts of studies is that the UI may not be optimized; in particular, for most AM the heat map covers the part of the image important for classification. Did the authors consider this or experiment with this?
> We agree that one major issue with attribution maps is they tend to highlight the main object regardless of whether the AI is correct or not (see L277). However, this is a general issue with attribution maps rather than a UI design choice.
> In fact, we had also tested with other attribution methods (e.g. VanillaGradient [7], IntegratedGradient [8], SHAP[9]) but excluded them from the study due to their noisy visualizations (which we found to have little to no utility).
>
> - [7] Simonyan et al.. Deep inside convolutional networks: Visualising image classification models and saliency maps. In ICML workshop 2014.
> - [8] Sundararajan et al. Axiomatic attribution for deep networks. ICML 2017.
> - [9] Lundberg et al. A unified approach to interpreting model predictions. NeurIPS 2017.
>
> ### > Additionally I believe there are some missing citations (other works which have evaluated explanation models): Chandrasekaran et al. EMNLP 2018. Hase et al. ACL 2020.
> Thank you very much for these relevant papers! We will cite them in the camera-ready version.
>
> ### > I wonder where participants were located and how the $10.20 compared to the minimum wage in that region.
> Our initial rate was USD 10.20 / hr, which is higher than the Prolific recommended rate wage of USD 9.60/hr. In fact, during the study, we had increased our rate to attract more participants (up to USD 13.68/hr). We will update the paper to reflect this.
> As participants come from various countries in the world (US, UK, Canada, India, Poland, Korea, South Africa, etc.), we did not consider minimum wage per region because this recommended rate is suggested by Prolific and accepted by all participants.

---

> > ### Author Response · Authors · 2021-09-01
> > **thank reviewer Tf4g! We hope to hear if you have further questions/thoughts**
> >
> > Dear reviewer **Tf4g**,
> >
> > Thank you for your inputs and encouragement again! We have happily updated our manuscript in light of your comments.
> >
> > We are a bit anxious and new to this NeurIPS 2021 author-reviewer discussion format---and we wonder if you have any further thoughts/questions given our rebuttal that we could address.
> >
> > Thank you very much!

---

### Decision · Program_Chairs · 2021-09-27

**Decision:**

Accept (Poster)

**Comment:**

This paper presents a user study aimed at evaluating the effectiveness of feature attribution methods (e.g. GradCAM) in assisting human decision making. Interestingly, feature attribution methods are found to not be better than a comparatively simple 3-nearest-neighbor-based method; in some cases, a feature attribution map was even harmful to human performance.

Reviewers found the paper to be compelling and well-written, though individual reviewers raised specific potential concerns, such as the size of the study (320 subjects) or the relative artificiality of the task (vs. a real-world application). However, in totality, reviews were positive and the authors’ rebuttals were helpful for clarifying reviewer questions.